# New evidence on the microstructural localization of sulfur & chlorine in polar ice cores with implications for impurity diffusion

Pascal Bohleber<sup>1,2,3</sup>, Nicolas Stoll<sup>3</sup>, Piers Larkman<sup>1,3</sup>, Rachael H. Rhodes<sup>4</sup> and David Clases<sup>5</sup>

<sup>1</sup>Alfred Wegener Institute Helmholtz Center for Polar and Marine Research, Bremerhaven, Germany.

<sup>2</sup>Goethe University Frankfurt am Main, Frankfurt am Main, Germany

<sup>3</sup>Ca' Foscari University of Venice, Department of Environmental Sciences, Informatics and Statistics, Italy.

<sup>4</sup>Department of Earth Sciences, University of Cambridge, Cambridge, UK.

<sup>5</sup>Nano Micro LAB, Institute of Chemistry, University of Graz, Graz, Austria

Correspondence to: Pascal Bohleber (pascal.bohleber@awi.de)

**Abstract.** Aerosol-related impurities play an important part in the set of paleoclimate proxies obtained from polar ice cores. However, in order to avoid misinterpretation, post-depositional changes need to be carefully assessed, especially in deep ice. Na, S and Cl are among the relatively abundant impurity species in polar ice (albeit still at the low ppb level in bulk samples), with important applications to paleoclimate reconstructions and dating, e.g. via identification of volcanic eruptions. Especially S has been studied intensely with respect to peak broadening with depth/age related to diffusion, but the precise physical mechanisms remain unclear. Mapping the two-dimensional impurity distribution in ice with laser ablation inductively coupled plasma mass spectrometry (LA-ICP-MS) has shown great potential for studying ice-impurity interactions, but the analytically more challenging elements S and Cl have not been targeted thus far. We show here that signals of S and Cl can be detected in Greenland and Antarctic ice by LA-ICP-MS mapping. In ice without evidence of volcanic activity, and unenhanced impurity concentrations, we obtain multi-elemental maps for Na, Cl and S at high resolution up to 10 µm and also include some exemplarily high resolution maps with a spot size down to 1 µm. We use Na as a previously investigated reference element and find a high level of localization of Na, S and Cl at grain boundaries but also some dispersed occurrence within grain interiors in dust-rich ice. The new maps support a view on diffusive transport not only through ice veins but also along grain boundaries. In the EPICA Dome C ice core samples we do not find any clear differences in impurity localization between samples from the Holocene and last glacial period. These results extend early studies targeting the localization of impurities, in particular through measuring S and Cl, and highlight the benefit of integrating such direct measurements with modelling efforts to determine the physical processes behind impurity diffusion.

#### 1 Introduction

Aerosol-related atmospheric impurity records are an important part of the unique combination of climate proxies archived in polar ice cores (e.g. Fischer et al., 2021; Legrand & Mayewski 1997). However, it is known that the original impurity distribution at the foundation of interpreted climatic signals can be disturbed by various post-depositional processes, and their

impact needs to be detected and constrained to avoid misinterpretation (e.g. Traversi et al., 2009, Stoll et al. 2021). Understanding the variability of impurities in the deepest and highly thinned ice core layers is particularly urgent for new "Oldest Ice" efforts to retrieve climatic records older than 1 million years from Antarctica – driven by the unsolved challenge to understand the cause of the "Mid-Pleistocene Transition" (Brook et al., 2006; Fischer et al., 2013; Yan et al., 2019; Wolff et al., 2022). Impurity (re-)localization is relevant for understanding climate record preservation, but it is also critical for the study of deformation processes on the microscale, and thus a topic of wide significance for ice core scientists and glaciologists alike.

Post-depositional relocation processes are expected to depend crucially on the physical and chemical properties of the individual impurity species. One very important process of post-depositional impurity relocation is the diffusion of ionic impurities through the ice matrix. Sodium, chloride and sulfate provide an important reference ensemble in this regard, motivating the focus of the work presented here. For the upper 350 m of the deep EPICA Dome C (EDC) ice core drilled in central Antarctica, Barnes et al. (2003) investigated the diffusion of ionic sodium, chloride and sulfate and found evidence for the diffusion of only chloride and sulfate. Evidence for diffusion came from the distinct broadening and damping of sulfuric acid peaks in meltwater analysis, associated with volcanic eruptions that provide crucial tie points for ice core chronologies (e.g., Severi et al., 2007; Svensson et al., 2013; Fujita et al., 2015; Sigl et al., 2022). The rate of sulfate diffusion is typically described by the term "effective diffusivity" (units: m<sup>2</sup> yr<sup>-1</sup>) - "effective" because it represents a time-weighted value that is the cumulative result of all diffusion experienced since deposition. Notably, the effective diffusivity metric lacks a foundational connection to a physical process (or processes) driving the diffusion in the ice. The localization of sulfur in the ice matrix is a key issue for describing diffusion processes in models, however. In a prominent modelling study, Rempel et al. (2001) suggested that a process termed "anomalous diffusion", happening within the connected network of veins, can lead to advection of sulfate peaks away from their original position. Ng (2021) achieved an important extension of the modelling efforts to include the "Gibbs Thompson effect", finding that if this effect is included, significant dampening of the peaks will prevent any detectable migration away from their original position. Regarding the implications of the findings, Ng distinguishes two scenarios, one with impurities completely localized at the ice veins (scenario 1) and another with significant contributions from grain boundaries and/or grain interiors (scenario 2). It becomes clear that the localization of impurities within the ice plays a key role in conceptualizing potential diffusion mechanisms. Yet, existing data do not provide sufficient constraint, e.g. to generally distinguish between the above two scenarios by Ng (2021). Notably the localization of the impurities in the ice can already provide significant new insight by itself and does not necessarily need to include precise concentration estimates.

Despite early attempts, observational evidence has not provided a fully comprehensive picture on the matter thus far. We describe a few key studies here; detailed summaries with additional information are given elsewhere (e.g. Stoll et al., 2021). Due to the low eutectic point of sulfuric acid, it was hypothesized that it is mostly located within a network formed by connected triple junctions (the intersection of grain boundaries of three grains) where it can diffuse along the veins

60

subsequently (Mulvaney et al., 1988). Barnes et al. (2003) discussed two different mechanisms distinguishing a "connected" versus "disconnected" ice vein or grain boundary network, both implying an indirect control on the effectivity diffusivity through the grain growth rate. Two recent studies extended this approach and focused on determining the effective diffusivity for sulfate at greater depths in the EDC core. Using the same input dataset, the determined values for the effective diffusivity differ beyond the Holocene, which is attributed mainly to different methodologies (Fudge et al., 2024; Rhodes et al., 2024). Importantly, the effective diffusivity of sulfate appears to decrease rapidly following deposition, with more than one order of magnitude difference between the early Holocene and the last glacial period (Rhodes et al., 2024). Rhodes et al. (2024) speculate that the decrease could be due to change in the localization of the impurities and therefore a change in diffusion mechanism.

Using cryo-scanning electron microscope (SEM) X-ray analysis or Raman spectroscopy on ice samples from shallow depths, S signals were detected primarily at a few investigated triple junctions, where very high concentrations were found but no trace of S and Cl was measurable elsewhere (Wolff et al., 1988; Mulvaney et al., 1988; Fukazawa et al., 1998). Later studies with cryo-SEM X-ray analysis found S mainly in inclusions but NaCl at grain boundaries (Cullen & Baker, 2000) and soluble impurities at grain boundaries (Barnes et al., 2002). Methodological differences, the low number of investigated triple junctions, the young age of the samples, and especially limits of detection have to be taken into account when attempting to put this work in a general view. At face value, Na and Cl are predominantly found at grain boundaries and triple junctions, while S may also be found in inclusions. Salts containing S (e.g. CaSO<sub>4</sub>) are primarily found in inclusions and the grain interior, respectively (Table 2 in Stoll et al., 2021), suggesting that their contribution to impurity diffusion is likely limited unless they become displaced to grain boundaries and possibly dissociate into soluble impurities.

85

Exploring the localization of impurities thoroughly over larger numbers of grains has so far been limited primarily due to methodological limitations. This particularly concerns detection limits hampering the collection of statistically significant large datasets that allow us to draw generalized conclusions. However, with the introduction of state-of-the-art impurity mapping techniques, laser ablation inductively-plasma mass spectrometry (LA-ICP-MS) has now become a new tool in investigating the localization of impurities in the ice matrix at tens of micron-resolution (Bohleber et al., 2020), extending previous 2D work with this technique (Della-Lunga et al., 2014). For mostly soluble elements such as Na, a high degree of localization at grain boundaries has been observed, while insoluble, dust-related elements can also be found in the grain interiors (Bohleber et al., 2023; Stoll et al., 2023). Initial studies in LA-ICP-MS ice core analysis and in particular impurity mapping have focused on analytically more easily accessible elements such as the metals Na, Mg, Al, Ca, Fe, Sr (Reinhardt et al., 2001; Della Lunga et al., 2014; Sneed et al., 2015; Bohleber et al., 2020; Hoffmann et al., 2024). The range of elements accessible to this technique has been steadily extended, in particular when using a "time-of-flight" mass spectrometer (LA-ICP-TOFMS) that in principle records the entire spectrum of elements (Wilhelms-Dick, 2008; Bohleber et al., 2023). Although the measurement of S and Cl is technically possible with this technique, they both come with their specific analytical challenges: For S, the mass of its primary isotope <sup>32</sup>S is heavily interfered by oxygen signals (<sup>16</sup>O<sup>16</sup>O<sup>+</sup>) and Cl suffers from low ionization efficiency in the

plasma. At the same time, sulfate, chloride and sodium belong to the relatively more abundant ionic impurity species in ice cores, although their concentrations in Antarctic ice typically remain in the low "parts per billion" (ppb) range when measured as bulk meltwater samples (e.g. Legrand & Mayewski, 1997). Recent evidence suggests that, due to the high degree of spatial localization at grain boundaries, concentrations of Na are substantially increased at the scale of tens of microns, which generally aids the detection and may bring S and Cl within the detectable concentration range of this technique (Bohleber et al., 2024).

Here, we report on recent work specifically targeting the mapping of S and Cl signals using LA-ICP-TOFMS within EDC ice core samples from the Holocene and last glacial period. Na serves as a reference element to compare our maps to previous LA-ICP-MS maps. The hypotheses put forward in recent studies served as the main motivation for this work: If the above-mentioned hypothesis by Rhodes et al. (2024) explaining the difference in effective diffusivity between the Holocene and the last glacial holds true, it would mean that impurity localization for S should be distinctly different in ice from these periods. If no such changes can be found, it would remain to be tested if evidence for changes in the vein connectivity with depth appears (Barnes et al., 2003). Similarly, based on mapping the localization of S it should become possible to distinguish between the two scenarios put forward by Ng (2021) – "ice vein only" versus "grain boundary and ice vein contributions". Note that localization is key to determining the plausibility of all these scenarios, and a fully quantitative analysis is not required. We thus focus on delivering information on the localization in the ice matrix rather than obtaining quantitative information on concentrations, which adds considerable complexity and uncertainty (Bohleber et al., 2024).

#### 2 Methods

#### 2.1 LA-ICP-TOFMS ice analysis

For the purpose of this study, two different systems were used for impurity mapping with LA-ICP-MS. In so doing we make use of an opportunity to discuss maps showing Na, Cl and S signals on ice from two initially unrelated measurement campaigns at the University of Graz and later at the Alfred Wegener Institute Helmholtz Centre for Polar and Marine Research, Bremerhaven (AWI). The newly established cryo-LA-ICP-TOFMS system at AWI consists of a 193 nm excimer laser ablation system capable of repetition rates up to 1 kHz (Iridia by Teledyne Photon Machines, Bozeman, MT, USA) coupled to a ICP-TOFMS (Vitesse ICP-TOFMS by Nu Instruments, Wrexham, UK). The latter includes a segmented reaction chamber, in which He and  $H_2$  are used at mL/min flow rates to remove spectral interferences. The laser ablation system is equipped with a cryogenic sample cell (CryoCOBALT Cell) using water-cooled Peltier elements to cool a sample holder. Two temperature sensors and a humidity sensor are used to control the cooling via an integrated software, ensuring that the sample surface temperature stays below -20°C. Temperature stability was better than +/- 1°C over more than 8 hours. An ARIS (Aerosol Rapid Introduction System, Teledyne Photon Machines, USA) is used for fast aerosol transfer to the ICP-TOFMS. Using a NIST612 glass standard, instrumental parameters are optimized for best sensitivity in the mass range from m/z = 23 to m/z=90

(comprising the mass range of primarily interesting analytes for ice and minimal oxides). NIST612 is used to tune the system for best single-pulse-response (SPR), including tuning the gas flows in the sample chamber as well as the distance between cell and sample surface. For these purposes, NIST glass standards were preferred over artificial ice standards due to easier use and higher homogeneity (Bohleber et al., 2024). Further details on the AWI setup can be found elsewhere (Bohleber et al., 2025).

As a second setup, we combined the cryogenic sample holder employed at Ca'Foscari University of Venice (Bohleber et al., 2020) with the LA-ICP-TOFMS system at the Institute of Chemistry, University of Graz (Niehaus et al., 2024; Lockwood et al., 2024). The latter comprises an Analyte G2 laser ablation system (Teledyne Photon Machines, USA) equipped with a HelEx II ablation chamber compatible with the Venice cryostage. The cryostage was operated in connection to a chiller circulating glycol-water mix cooled to -27°C, achieving sufficient cooling of the sample with the temperature at the surface measuring below -20°C and with no visual evidence of melting. Again, this ablation chamber is connected with an ARIS rapid aerosol introduction system to an ICP-TOFMS (Vitesse ICP-TOF-MS by Nu Instruments, Wrexham, UK). Since using an identical ICP-TOFMS, the two systems are complementary through the differences in the laser ablation instrument. The Graz system was used primarily with spot sizes of 20 and 40 µm over comparatively large areas, while the AWI system was used for obtaining smaller maps at higher resolution analysis at 10 µm spot size. Table 1 summarizes the settings used by both LA-ICP-TOFMS systems.

Table 1: Overview on instrumental settings used for LA-ICP-TOFMS ice core analysis AWI

|            | ICP-TOFMS (Vitesse)                                      |                                                                                                                                                                                                                                                                                                                                                                                  |
|------------|----------------------------------------------------------|----------------------------------------------------------------------------------------------------------------------------------------------------------------------------------------------------------------------------------------------------------------------------------------------------------------------------------------------------------------------------------|
| 193        | RF power (W)                                             | 1300                                                                                                                                                                                                                                                                                                                                                                             |
| 4          | Auxiliary gas flow (L min <sup>-1</sup> )                | 2                                                                                                                                                                                                                                                                                                                                                                                |
| 500 / 1000 | Coolant flow (L min <sup>-1</sup> )                      | 13                                                                                                                                                                                                                                                                                                                                                                               |
| 10         | Nebulizer flow (L min <sup>-1</sup> )                    | 1.25                                                                                                                                                                                                                                                                                                                                                                             |
| 1,5,10     | Reaction cell gas (mL min <sup>-1</sup> )                | 7 (He) / 12 (H <sub>2</sub> )                                                                                                                                                                                                                                                                                                                                                    |
|            |                                                          |                                                                                                                                                                                                                                                                                                                                                                                  |
|            | ICP-TOFMS (Vitesse)                                      |                                                                                                                                                                                                                                                                                                                                                                                  |
| 193        | RF power (W)                                             | 1450                                                                                                                                                                                                                                                                                                                                                                             |
| 4          | Auxiliary gas flow (L min <sup>-1</sup> )                | 2                                                                                                                                                                                                                                                                                                                                                                                |
| 300        | Coolant flow (L min <sup>-1</sup> )                      | 13                                                                                                                                                                                                                                                                                                                                                                               |
| 10         | Nebulizer flow (L min <sup>-1</sup> )                    | 1.35                                                                                                                                                                                                                                                                                                                                                                             |
| 20, 40     | Reaction cell gas (mL min <sup>-1</sup> )                | 14 (He) / 6 (H <sub>2</sub> )                                                                                                                                                                                                                                                                                                                                                    |
|            | 4<br>500 / 1000<br>10<br>1,5,10<br>193<br>4<br>300<br>10 | 193 RF power (W)  4 Auxiliary gas flow (L min <sup>-1</sup> )  500 / 1000 Coolant flow (L min <sup>-1</sup> )  10 Nebulizer flow (L min <sup>-1</sup> )  1,5,10 Reaction cell gas (mL min <sup>-1</sup> )  ICP-TOFMS (Vitesse)  193 RF power (W)  4 Auxiliary gas flow (L min <sup>-1</sup> )  300 Coolant flow (L min <sup>-1</sup> )  10 Nebulizer flow (L min <sup>-1</sup> ) |

We focused here on measuring maps of the following isotopes:  $^{35}$ Cl,  $^{37}$ Cl and  $^{32}$ S,  $^{34}$ S, with  $^{23}$ Na used as a reference element with respect to previous studies. The strategy mainly employed for the detection of Cl and S focused on achieving maximum sensitivity in the lower mass range, by specifically tuning the ICP-TOFMS for high sensitivity in the range of m/z=23 to 90. In doing so, we also carefully tuned parameters of the segmented reaction chamber, including the cell gas flows of He and H2 in order to maximize the breaking up of  $^{16}$ O $^{16}$ O $^{+}$  while limiting the amount of sensitivity decrease to  $^{32}$ S. An additional approach was to reduce the amount of background O coming from the ablation chamber via sublimation of the ice surface, i.e. through the constant background stream of H2O to the ICP-TOFMS. For this purpose, at the University of Graz, we successfully explored lowering the gas flows inside the ablation chamber, while respecting the limits imposed on this to not negatively affect SPR. With the AWI setup, the ablation chamber has a built-in humidity sensor, which shows that the humidity can be decreased by running additional gas evacuation and refilling cycles. We typically used 20-25 such cycles initially and also applied about 30 minutes of constant flushing with dry He. Thereby, the humidity was found to decrease rapidly and then remain at typically around 5-10%. Further details on the analytical challenges of measuring S and Cl are discussed in Sect. 4.1.

# 2.2 Sample selection



We chose a set of ice core samples both from the Holocene and the last glacial period in the EDC ice core. In order to test the mapping of S and Cl on a broad range of ice conditions and impurity concentrations we also included one sample of the East Greenland ice-core project (EGRIP) core drilled in East Greenland. The glacial EGRIP sample (EGRIP 2286) contains a dustrich cloudy band from the Younger Dryas and thus represents contrasting conditions to the low impurity samples from EDC. It contains insoluble particle concentrations > 250,000 particles/ml (Stoll et al., 2022). This sample selection deliberately excluded ice containing known traces of volcanic eruptions, in order to provide a realistic subsample of typical polar ice conditions. We name subsamples with a hypen to denote their sequence below the top of the bag number, for example subsamples denoted as EDC 1819-2, 1819-3 etc are taken from the respective bag (1819) and are of 8 cm length, i.e. originating from 8 and 16 cm below the top of the bag, respectively. The same applies to EDC 513-2. Table 2 provides an overview, including information on the respective age of the sample. As a general reference for bulk ionic Na<sup>+</sup> and (non-sea-salt) SO<sub>4</sub><sup>2-</sup> concentrations in EDC we report in Table 2 data for the depths 280.5, 584.1, 999.9 m (Wolff et al., 2006).

Table 2: Overview on ice samples analyzed in this study. The analysis in Graz produced maps larger in size at comparatively coarser spot sizes while the analysis at AWI focused on finer resolution over smaller maps. Bulk ionic values are reported for EDC from Wolff et al. (2006) for adjacent depths (see text).

| Core name | Bag n° | Depth | Age     | Period       | Resolution          | <b>Bulk concentration</b> |
|-----------|--------|-------|---------|--------------|---------------------|---------------------------|
|           |        | [m]   | [ka BP] |              | beam size [µm]      | Na / Cl/ SO4 (ppb)        |
| EDC       | 513    | 281.6 | 9.0     | Holocene     | 10 (AWI), 40 (Graz) | 20 / 30/ 85               |
| EDC       | 1065   | 585.2 | 27.3    | Last glacial | 20 (Graz)           | 74 / 106 / 148            |
| EDC       | 1819   | 1000  | 64      | Last glacial | 10 (AWI)            | 90 / 112 / 212            |

EGRIP 2286 1256.9 12.1 Younger Dryas 20 (Graz)

Following our previously established approach for impurity mapping (Bohleber et al., 2020), the ice surface was decontaminated by scraping with a major-element free ceramic ZrO<sub>2</sub> blade (American Cutting Edge, USA) immediately before inserting the samples into the ablation chamber. Prior to obtaining maps on the ice, the measured area was also cleaned by preablation. Mosaics of optical images of the ice surface were recorded using the camera integrated with the laser ablation systems. Maps were acquired as adjacent horizontal lines covering a rectangular area (Bohleber et al., 2020). The maps were generated using HDIP (Teledyne Photon Machines).

#### 3 Results





Experimental constraints in measurement time and resources require making a trade-off between the size of the mapped area and the resolution (spot size). Large size maps at comparatively coarse resolution (40 and 20 µm) typically offer a good overview on the degree of impurity localization at grain boundaries versus grain interiors. Finer resolution (10 µm and finer) may be required to investigate impurity variability within grain boundaries and in particular triple junctions. The result section is structured accordingly, going from coarser resolution large size maps to finer resolution detailed maps taking a closer look at grain boundaries and triple junctions.

# 3.1 Large size maps

Figures 1 and 2 show examples for the large size 32S intensity maps alongside 37Cl and 23Na recorded with the Graz LA-ICP-TOFMS system. Additional maps are included in the Supplementary Material. As mentioned in the introduction, Na primarily serves as a reference element showing the high degree of localization at grain boundaries, analogous to previous observations in EDC and EGRIP (Bohleber et al., 2020; Bohleber et al., 2023; Stoll et al., 2023). The maps of S reveal a similar picture as for Na, with high localization at grain boundaries and low intensities within the grain interiors. In EGRIP 2286, a sample containing a dust-rich cloudy band from the Younger Dryas, some isolated pixels within the grain interior also show high S values (Fig. 2). The EGRIP 2286 cloudy band sample likely has significantly higher impurity content compared to the EDC 513 Holocene sample, reflected in the higher signal amplitude in spite of less material ablated with the smaller spot size used (20 µm EGRIP 2286 vs 40 µm EDC 513). Both maps show intensity variability along the grain boundaries, indicating that some spatial variability in grain boundary concentrations exists on the scale of a few mm. Note for example lower intensities highlighted in the upper right area of the EGRIP 2286 map (Fig. 2) and higher relative intensities in the lower left of the EDC 513 map (Fig. 1). At 40 µm and 20 µm spot size, respectively, no clear evidence for enhanced concentrations at triple junctions relative to grain boundaries is detected, e.g. via pixels with brighter intensity at triple junctions. For a more quantitative evaluation of this finding, pixels belonging to grain boundaries (including triple junctions) were segmented from the Na and S maps using a watershed algorithm in HDIP analogously to previous work (Bohleber et al., 2023). Pixels representing triple junctions were additionally segmented out manually. Albeit systematically above the median of the grain boundary intensities, the median of the triple junctions falls within the 25-75% interquartile range of the grain boundary intensities. The results are shown in Fig. S4 in the Supplement together with the intensity histograms of the Na and S maps of Fig. 1 and Fig. 2.

Figure 1: Map of EDC bag 513 (40 μm spot, 6.4 x 10.4 mm). Shown are intensities in counts (cts) for <sup>23</sup>Na (top left), <sup>32</sup>S (bottom left) and <sup>37</sup>Cl (bottom right) together with the optical mosaic (top right).

Figure 2: Map of EGRIP bag 2286 (20  $\mu$ m spot, 4 x 17 mm). Shown are intensities in counts (cts) for <sup>23</sup>Na (top row), <sup>32</sup>S and <sup>37</sup>Cl (middle rows) and the optical mosaic (bottom)

#### 220 3.2 High-resolution maps obtained with 10 μm spot size


Figures 3-5 show the results for EDC 1819-2, 1819-3, and 513-1 respectively. An additional map was recorded on EDC1819-3 in one session right after the map on EDC513-1 (shown in the Supplementary Material). The maps show that also at  $10 \mu m$  spot size, Na, S and Cl are highly localized at grain boundaries. Notably these maps also show signal distributed among different isotopic channels. The map for  $^{35}$ Cl contains only noise in all maps recorded (hence not shown), but strong signal in  $^{37}$ Cl, which indicates the potential formation of Chlor- hydrogen adducts (see Sect. 4.1). As shown in Fig. 3-5 for S,  $^{32}$ S

contains most of the signal. Despite the low isotopic abundance, some maps even reveal signal in the isotopes <sup>33</sup>S (on rare occasion) and <sup>34</sup>S. Also, at 10 µm spot size, there is no clear evidence for enhanced intensities at or around triple junctions.

Figure 3: Example map for EDC1819-2, 10 μm spot size. Row a) shows the optical mosaic with dark lines denoting grain boundaries and the map for <sup>23</sup>Na. Note the potential discontinuity in the grain boundary intensity marked by a red square; the same area is shown as a zoom-in section. White and yellow arrows indicate examples of an intra-grain intensity spot for Na only and Na with S, respectively. Row b) contains the maps for the two main channels with signals for S and Cl, <sup>32</sup>S and <sup>37</sup>Cl respectively. Under best case mapping performance, other isotopes of sulfur may also show signals, illustrated in row c). The white horizontal line on the right-hand side in the <sup>37</sup>Cl and <sup>34</sup>S maps is an absence of signal as a result of an autoblanking procedure protecting the detector in the ICP-TOFMS.

Figure 4: Example map for EDC1819-3, 10µm spot size. The channels for the isotopes <sup>33</sup>S and <sup>35</sup>Cl contain noise only and are therefore not plotted. Shown on the top is the optical mosaic. Note the potential discontinuity in the grain boundary intensity marked by a red square in the <sup>23</sup>Na map and the respective zoom-in on the top left.

Figure 5: Example map for EDC513-1,  $10\mu m$  spot size. The optical mosaic is shown in the top left. Features are visible in each plotted S channel, while Cl remained confined to m/z=37 (see text).

# 3.3 Ultra-high resolution maps obtained with 1 µm spot size

Based on the limited evidence for enhanced impurity localization at triple junctions in maps collected with spot sizes 40, 20 and 10 μm, an exemplary investigation was performed with even smaller spot sizes. For this purpose, a triple junction was mapped with subsequently finer resolution, with 10 μm, 5 μm and 1 μm spots (Fig. 6). The results for Na show that at 10 and 5 μm, the triple junction is confined to a single pixel, whereas the maps recorded with 1 μm show more detail and a region of brighter intensity at the triple junction. Due to the much smaller volume ablated, it was not possible to obtain signals for S, although some signal was detectable for <sup>37</sup>Cl.

Figure 6: Exemplary investigation into the influence of spot size on detecting enhanced localization at triple junctions versus grain boundaries. The top row shows ultra-high resolution maps of a triple junction recorded at 1 µm spot size, alongside an optical mosaic of the same area. Note that the lower intensity on the right-hand side grain boundary is likely impacted by the scanning-direction, an effect only noticed at this high resolution (see text). The bottom row shows Na maps recorded over the same triple junction with 5 and 10 µm spot size.

#### 4 Discussion

255

260

#### 4.1 The analytical challenge of detecting sulfur and chlorine in ice with LA-ICP-TOFMS

As demonstrated by the results shown in Fig. 1 – 6, high-resolution mapping of both S and Cl is possible in ice with LA-ICP-TOFMS but come with their own specific challenges. The main hurdle for detecting S is the mass interference by  $^{16}O^{16}O^{+}$  on m/z=32, which results in high background levels. As opposed to the analysis of other materials using dry plasma conditions, the  $^{16}O^{16}O^{+}$  background cannot be avoided for ice since it is caused to a large degree by ablating the ice matrix itself, resulting

in a constant stream of H<sub>2</sub>O to the ICP-MS. Mitigation strategies involve i) high mass resolution in order to resolve the closely-spaced two peaks the mass spectrum caused by <sup>16</sup>O<sup>16</sup>O<sup>+</sup> and <sup>32</sup>S, ii) breaking up <sup>16</sup>O<sup>16</sup>O<sup>+</sup> by collision or reaction in the segmented reaction cell, iii) mass shifting of <sup>32</sup>S through chemical reaction, e.g. with Oxygen. Strategy iii) is S-specific and typically used in applications with a triple-quadrupole instruments (Martinez-Sierra et al., 2015), whereas i) and ii) can be applied in LA-ICP-TOFMS also without an additional quadrupole mass filter. In a previous pilot study, we were also able to resolve signals of <sup>32</sup>S using a Teledyne Photon Machines G2 laser ablation system (as used in Graz) coupled to an icpTOF 2R (TOFWERK, Thun, Switzerland) (Bohleber et al., 2021). Regarding the detection of Cl, an important side-effect of using the segmented reaction chamber is the formation of Chlor-hydrogen adducts, which explains the lack of signal in the mass channel m/z=35 in our maps (Figure S5 in the Supplementary Material). This channel should contain the majority of the Cl signal based on isotope abundance (about 75% for <sup>35</sup>Cl and 25% for <sup>37</sup>Cl). However, by monitoring <sup>35</sup>Cl and <sup>37</sup>Cl whilst analyzing Cl standards, we found conclusive evidence that the entirety of <sup>35</sup>Cl signal is mass-shifted to <sup>37</sup>Cl by formation of H<sub>2</sub>Cl<sup>+</sup>. It is likely that also <sup>37</sup>Cl is mass-shifted and detected at <sup>39</sup>K instead.

Ultimately, the strategy outlined above, requiring specific tuning of the ICP-TOFMS and controlling the background levels of Oxygen add considerable complexity to the already elaborate nature of high-resolution impurity mapping in ice cores. Future efforts will require making a tradeoff between achieving an adequate degree of sample throughput versus such time-consuming fine-tuned analysis. The latter could be potentially further extended in quantitative analyses through dedicated artificial ice standards. Since not required for this study (see introduction) and due to the already complex measurement strategy, we have not added the quantitative dimension considering the substantial complexity and added uncertainty involved in the use of artificial ice standards (Bohleber et al., 2024). Further extensions to this work for the detection of S could include exploring the use of alternative collision-reaction cell gases, such as Xe or CH<sub>4</sub> (Guillong et al., 2008; Singh et al., 2025).

The maps showing a signal not just in <sup>32</sup>S but also in the much less abundant (but less interfered) isotopes <sup>33</sup>S and <sup>34</sup>S also motivate exploring in more detail the limits to isotopic analysis with LA-ICP-TOFMS. This could involve specific sections with known volcanic eruption, for which the added experimental effort would be warranted. If taken at face value, the relative abundance of the sulfur isotopes agrees within their uncertainty (typically less than 3%) with the natural isotopic abundance, 95, 0.75 and 4% for <sup>32</sup>S, <sup>33</sup>S and <sup>34</sup>S, respectively (Meija et al, 2016). We calculated for this purpose the mean intensity at grain boundaries (the location of the signal in the maps) in Fig. 1 and 5 for all three isotopic channels (Table 3).

Table 3: Average of intensities (in cts) at grain boundaries calculated for sulfur isotope channels. The maximal uncertainty for the relative abundance values is 3%.

| Sample    | Isotope | Average  | <b>Uncertainty (1s)</b> | Relative Abundance (%) |
|-----------|---------|----------|-------------------------|------------------------|
| EDC1819-2 | 32S     | 21300000 | 330000                  | 94                     |
|           | 33S     | 208000   | 3700                    | 1                      |

|          | 34S | 1240000  | 15000  | 5  |
|----------|-----|----------|--------|----|
| EDC513-1 | 32S | 12100000 | 140000 | 93 |
|          | 33S | 134000   | 2600   | 1  |
|          | 34S | 717000   | 8300   | 6  |

# 4.2 Implications for diffusion studies

Due to the sophisticated nature of LA-ICP-MS impurity mapping in general and mapping S and Cl in particular, the amount of data remains limited. The presented maps have to be regarded as snapshots covering comparatively small areas. Although some spatial variability in the levels of S and Cl at grain boundaries is detected by the larger maps (Fig. 1 and 2), the localization of S and Cl at grain boundaries and triple junctions is found consistently in all maps. Thus, this has to be considered a robust result.




While methodological differences have to be kept in mind, our data are consistent with previous studies investigating S e.g. by cryo-SEM and X-ray analysis or Raman spectroscopy, finding S signals in a limited number of samples and measurements primarily at triple junctions, potentially indicating liquid-filled veins (e.g. Mulvaney et al., 1988; Fukazawa et al., 1998). Regarding a crucial point in conceptualizing impurity diffusion along ice veins, the question of a potential enrichment of impurities at triple junctions is much harder to answer based on our data. The exploration of a triple junction at ultra-high resolution (1  $\mu$ m) points towards spatial resolution being a key issue in this context: only at this fine scale can the intensity difference be clearly recognized in our example (Fig. 6). At 1  $\mu$ m, the investigated triple junction does show a local increase in intensity by about a factor of 1.3 relative to the grain boundary at the lower left. This grain boundary was chosen because the intensity of the other two grain boundaries (at the top and to the lower right of the map) was found to depend on the scan direction used to acquire the map (horizontal versus vertical scans, Fig. S3). This effect was only observed to occur at 1  $\mu$ m spot size and illustrates another dimension of complexity in impurity mapping at this high resolution. Adding to this are the higher limits of detection due to the much smaller amount of ablated material ablated at 1  $\mu$ m (hence no S signal in the maps) and the higher amount of resources consumed (almost 200k shots were needed for the small map in Fig. 6).

The primary bottleneck in terms of resources is measurement time, which also translates into consumption of energy and cell gases, however. As a consequence, we only investigated mapping at 1 µm resolution exemplarily in this work. Yet, dedicated mapping of triple junctions at 1 µm resolution appears feasible for future studies mastering some methodological challenges such as the scan direction influence and resource consumption. Computer-vision assisted mapping may provide a better compromise between resolution and resource consumption in this regard (Larkman et al., 2025a). It is also worthwhile mentioning that the apparent spatial extent of the grain boundaries and triple junctions in the maps can change depending on the applied resolution. At 10 µm spot size, grain boundaries typically extend over 2-3 pixels in width, for 5 µm this reduces to 1-2 pixels. An identical dosage of 10 was used for all maps. For 1 µm, the spatial extent is around 5 pixels, roughly consistent

with the 5 μm maps. Notably, this only refers to the apparent size of surface features in the chemical maps, which has been influenced by sublimation. The 1 μm spot size may be more capable of resolving the fine-scale impurity variability around triple junctions, and shows that some enhancement of Na at triple junctions relative to grain boundaries may exist at that spatial scale. Mulvaney et al., 1988 applied a comparable spatial resolution of around 1x1 μm in their SEM analysis supporting our results. Also in this SEM analysis, such a local enhancement at triple junctions could have pushed signals to the detectable range, while signals at grain boundaries remained undetectable.






At this point, our results do not support the notion of vein-dissolved ionic impurities dominating the respective bulk ice core records (e.g. referred to as "scenario 1" by Ng (2021)). Similar to Na, also S and Cl appear to be highly localized at grain boundaries, with a minor enhancement possible in veins at triple junctions. The fact that grain boundaries are populated to a similar degree as triple junctions suggests that diffusion of soluble impurities in ice needs to be treated considering grain boundary surfaces in three dimensions. Here, changes in grain size, e.g. via recrystallization and grain growth, becomes a crucial factor (Ng, 2021). The localization of S and Cl at grain boundaries has significant implications for the interpretation of 1D LA-ICP-MS profiles designed to collect climate-related signals. The impurity-boundary imprint manifests strongly in such measurements and may also become relevant for bulk meltwater measurements, such as with continuous flow analysis (CFA), particularly in ice samples with large average grain sizes (Larkman et al., 2025b).

It has to be noted in this context, however, that dust-rich ice such as the EGRIP 2286 cloudy band sample has already shown higher concentrations of Na also in grain interiors (Fig. 2 and Bohleber et al., 2023), similar to other EGRIP cloudy band samples (Stoll et al., 2023). Some cases with Na less abundant in grain interiors are also present in the glacial EDC samples of bag 1819 (Fig. 3 and 4). It is thus likely that S in dust-rich samples is also more prominent in the grain interior. On these grounds, we tentatively note that S and Na appear also in isolated clusters in the grain interior (although some co-localization of S with Na also occurs, e.g. Fig. 3), while Cl is often co-localized with Na (likely indicating NaCl). The fact that some clusters in grain interiors are found in the maps is consistent with evidence for sulfate-bearing micro-inclusions such as salts found in grain interiors (e.g. Ohno et al. 2005, Eichler et al., 2019, Stoll et al., 2022). Notably, inter-method comparisons are always a challenge and LA-ICP-MS cannot directly distinguish dissolved and particulate fractions, other than identifying particles via spatial information on their localization (isolated clustered pixels of high intensity in grain interiors) (Bohleber et al., 2023). In a simplified view, however, depending on the degree to which salts can dissociate and be displaced to grain boundaries due to e.g., grain growth, they could contribute to the overall apparent impurity mobility via diffusion. Similarly, dissolution of minerals resulting from chemical reactions at grain boundaries is a possibility (Baccolo et al., 2021). Accordingly, the physical reality likely is a mix between the two scenarios described in Ng (2021). For our "snapshots" we find a high degree of localization at grain boundaries for Na. S and Cl. with only minor contributions from the grain interiors. However, in general the relative contributions from ice veins, grain boundaries and grain interiors can depend on conditions such as climate period and insoluble particle content, and uncharacterized microstructural processes.

Notably, the situation in deep ice sections, with larger grains and more time spent at higher temperatures, may deviate from the snapshots from relatively shallow depths presented here and remains to be explored, in particular regarding the influence of geochemical reactions leading to dissolution at grain boundaries (Baccolo et al., 2021). Sections with high S-peaks from volcanic eruptions, which have not been measured here, may also prove to be a special case. At least for sections within the Holocene and last glacial period in the EDC core, it is difficult to conceive systematic differences between the data presented here, however, other than the bulk S concentration levels.


390

In considering a physical process to explain their empirical constraints on the effective diffusivity of sulfate, based on data from EDC Holocene ice. Barnes et al. (2003) invoked either the diffusion through a connected vein network or a system of discontinuous veins with interconnectivity changes due to grain growth. In recent studies, both Fudge et al. (2024) and Rhodes et al. (2024) find an order of magnitude lower values for the effective diffusivity beyond the Holocene. Possible explanations of the lower effective diffusivity involve i) sulfuric acid not readily diffusing in liquidlike veins (Fudge et al., 2024), ii) changes to an initially inter-connected network of veins and iii) changes in the mechanism of diffusion resulting due to sulfate ions located within the ice microstructure (Rhodes et al., 2024). Diffusion both through veins and grain boundaries appears to be supported by the maps obtained in the present work, while a change to sulfate predominantly located in the grain interiors is not. This means the new maps do not indicate changes in the general character of impurity localization between Holocene and glacial samples, which would support the notion of a change in diffusion mechanism according to (iii). In contrast, the new 375 data are consistent with a recently formulated view that the majority of the diffusion is triggered at shallow depths and that only minor differences occur with depth (Ng et al., 2025). It is important to note that, based on the inevitable snapshot character of the data, it is not possible to make generalized statements, in particular regarding the level of 3D interconnectivity of veins and grain boundaries. As indicated in Fig. 3 and 4, we find some occasional evidence of partially interrupted impurity population at grain boundaries in the glacial maps but not in the Holocene. Notably, the discontinuities among grain boundaries 380 in Fig. 1 are due to air bubbles, which is not the case for Fig. 3 and 4. However, the discontinuities in the impurity population at grain boundaries are rare, and do not concern the triple junctions. This does not support the potential explanations i) nor ii) above, and points towards a potential additional mechanism at work. An investigation taking into account 3D geometric effects and grain size changes may provide further insight. A possible approach to investigate the 3D interconnectivity would be a LA-ICP-MS high resolution study ablating layer by layer subsequently resulting in quasi-3D mapping. The large amount of 385 time and resources needed hamper the practicality of this approach so far. Additional exemplary investigations could be in reach soon, however, similar to the 1 µm mapping performed here. Further semi-empirical investigations could employ a combination of LA-ICP-MS maps with 3D modeling of the grain-scale chemistry (Larkman et al., 2025b; Ng 2024; Ng 2021).

Interestingly, Barnes et al. (2003) found evidence for signal damping only for Cl and S, but not for Na, suggesting that a diffusive process transported sulfate and chloride ions but sodium ions remained fixed. More precisely constraining the

mobility of Cl would also be important for cosmogenic radionuclide dating with <sup>36</sup>Cl (e.g. Muscheler et al., 2005). The maps collected here provide no support for the notion that S and Cl are more mobile than Na. For the Holocene and glacial maps, which were obtained sequentially in one session (Fig. 5 and Fig. S2), and hence not significantly affected by instrumental drift, we compared the average intensity ratios of S/Na and Cl/Na at grain boundaries but found similar values for the Holocene and glacial samples (although notably, the ratio is strongly influenced if spurious low Na intensity pixels from the grain interiors are included). Tentatively, this result argues against a relative difference, e.g. glacial ice being more depleted in potentially more "diffusive" elements such as S and Cl versus Holocene samples. On the other hand, the co-localization of Na and Cl at grain boundaries could entail processes slowing of diffusion, e.g. by anion-cation trapping decreasing the mobility by electrostatic attraction and eventually leading to the formation of NaCl. This effect was observed in laboratory experiments decreasing the mobility of HCl in ice and was suspected also for other ionic species (Livingston & George, 2001). Since LA-ICP-MS cannot directly distinguish particulates (e.g. salt formation) and dissolved impurity fractions, a combined approach e.g. with cryo-Raman spectroscopy (albeit not for NaCl in ice) may help to shed light on potential chemical reactions taking place within grain boundaries. This would provide important new insight into geochemical reactions in ice and ultimately aid an improved signal interpretation, especially in deep ice conditions such as in the "Oldest Ice".

#### 405 Conclusions

The investigation of impurity localization and ice-impurity interactions has gained new momentum with the introduction of state-of-the-art impurity mapping using LA-ICP-MS. So far, the elemental range in focus with this method has not been extended to the analytically more challenging elements S and Cl. Both are relatively abundant in polar ice and are of significance for paleoclimate reconstructions and dating efforts, but are also known to be affected by post-depositional processes, such as diffusion and chemical reactions. As part of the ongoing progress in advancing LA-ICP-MS, we show here that signals of S and Cl can be detected in Greenland and Antarctic ice using LA-ICP-TOFMS mapping, albeit requiring careful fine-tuning of the instruments. In doing so, it becomes possible to obtain multi-elemental maps at high resolution, up to 10 µm, and, exemplarily even at 1 µm. We observe a high level of localization of S and Cl at grain boundaries, which has already been observed for other elements such as Na. Additionally, there is a dispersed occurrence within grain interiors in dust-rich ice. Evidence for enhancement at triple junctions, and hence in ice veins, is limited to the ultra-high resolution map at 1 µm. The new maps support a view on diffusive transport not only through ice veins but also grain boundaries, but do not show any clear differences in this regard between samples from the Holocene and last glacial period in the EDC ice core. To determine the physical processes behind impurity diffusion, our results emphasize the merit of integrating such direct measurements with studies on empirical evidence and modelling. Such a combined approach promises a new level of insight into impurity signal preservation in polar ice cores.

### Data availability

The underlying data will be made available via a public data repository (e.g. Pangaea) after completion of the peer-review process.

#### **Author contributions**

Experimental design was developed by PB and DC. Measurements were conducted by PB, NS, PL and DC. RR contributed expertise on impurity diffusion. All authors contributed to the discussion of the results and the final version of the manuscript.

#### **Competing interests**

The authors declare that they have no conflict of interest.

#### Acknowledgements

We are grateful to Remi Dallmayr, Raquel Gonzalez de Vega, Lukas Schlatt & Ciprian Stremtan for their valuable input and help in the laboratory. Pascal Bohleber gratefully acknowledges funding from the European Union's Horizon 2020 research and innovation program under the Marie Skłodowska-Curie grant agreement no. 101018266. Nicolas Stoll and Pascal Bohleber gratefully acknowledge funding from the Programma di Ricerche in Artico (PRA). Nicolas Stoll further acknowledges funding from the European Union's Horizon 2020 research and innovation program under the Marie Skłodowska-Curie grant agreement no. 101146092. Co-funded by the European Union (ERC, AiCE, 101088125 and NanoArchive, 101165171). Views and opinions expressed are however those of the authors only and do not necessarily reflect those of the European Union or the European Research Council. Neither the European Union nor the granting authority can be held responsible for them. We also thank Hanna Brooks, an anonymous referee and editor Kaitlin Keegan for their help in improving the manuscript.

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
