# Peer review of "New evidence on the microstructural localization of sulfur & chlorine in polar ice cores with implications for impurity diffusion"

_EGUsphere, 2025_

## Referee Comment (RC1)

**Review of Bohleber et al. manuscript #egusphere-2025-355**

**"New evidence on the microstructural localization of sulfur, chlorine & sodium in polar ice cores with implications for impurity diffusion"**

**Overall quality of the preprint manuscript**

This preprint manuscript presents new findings on the spatial distribution of sulfur (S), chlorine (Cl), and sodium (Na) in polar ice using high-resolution laser ablation inductively coupled plasma mass spectrometry (LA-ICP-TOFMS). The study focuses on samples from the EPICA Dome C (EDC) core in Antarctica and the EGRIP core in Greenland, aiming to understand impurity localization and its impact on post-depositional diffusion. The results show that S, Cl, and Na are predominantly localized along grain boundaries, with limited evidence for accumulation at triple junctions or within grain interiors, except in dust-rich ice. These findings suggest that diffusion may occur primarily along grain boundaries rather than through interconnected veins, challenging assumptions that post-depositional mobility varies significantly with climatic periods. The study highlights the need to integrate impurity mapping with modeling efforts to better constrain diffusion processes that affect paleoclimate records preserved in deep ice cores.

While the fundamental science discussed in the preprint is a significant contribution to the field and is clearly within the scope of The Cryosphere, the manuscript text requires substantial reworking before it is suitable for publication. In particular, the framework established in the introduction and methods sections lacks clarity and cohesion, making the central aims and motivations of the study difficult to follow. This undermines the accessibility of the results and their implications, even for readers familiar with ice core and/or LA-ICP-MS science. Once the manuscript has undergone major revision, it will be a lovely addition to The Cryosphere.

**Individual scientific questions/issues/comments**

For ease of reading, specific comments, questions, and issues have been broken into sections. Generally for each section, a list of small, specific comments by line number follows the more general comments.

**Abstract**

The abstract does a great job of summarizing the work presented. There are just a few comments to help readers as they are introduced to the presented work.

**Specific comments by line number:**

- **Line 12:** "Na, S and Cl are among the relatively abundant impurity ... precise physical mechanisms remain unclear."

  Does one expect that similar post depositional effects are seen in Na and Cl? Do we know enough about the system to know that there is an element that does not suffer post-depositional changes?

- **Line 15:** "Mapping the two-dimensional impurity distribution ... S and Cl have not been targeted thus far."

  Can you give us an example of what elements work well?

- **Line 17:** "We show here that signals of S and Cl can be detected in ... exemplarily even 1µm."

  The grammar at the end of the sentence ("exemplarily even 1µm.") is really awkward. Please rewrite for clarity.

- **Line 19-20:** "We find a high level of localization of S and Cl (and Na) at grain boundaries but also some dispersed occurrence within grain interiors in dust-rich ice."

  Why is Na pulled out into parentheses? Is this a paper about S, Cl and Na? Or about S and Cl?

- **Line 20-23:** "The new maps support a view on diffusive transport not only through ice veins but also along grain boundaries, but do not show any clear differences in this regard between samples from the Holocene and last glacial period in the EDC ice core."

  This sentence is long and bulky. Consider splitting it into two. Also you have not defined EDC yet, please do so here. The casual reader, even someone who works in the field will likely not know what you are talking about.

**Introduction**

While the introduction does introduce concepts relevant to the manuscript and science, the structure of the introduction is disjointed and the message muddled. The majority of the introduction is dedicated to understanding a single element, S. Small interjections mention Cl and Na as well, but the reader is left confused as to why we are looking at three elements, if only one is of importance. There are also large leaps in text, rather than presenting a cohesive narrative.

It seems best to take a step back and outline the key information that needs to be conveyed. Once a clear flow of information has been outlined, the narrative will quickly come together. Most of the pieces are already there. Perhaps something like this?

- First, talk about the fact that we work with a fundamental assumption that there is no post-depositional migration of species. However, we know that this assumption is a simplification. Here are some post-depositional effects that we are really concerned about. Critically, ice cores that have much to tell us about the very distant past are at the highest risk of experiencing these migrations. Specifically, the community knows that species $X_i$ $Y_i$ Z are of particular concern. There is a high level of concern about S because we know that it migrates at depth and is one of the underpinning species for age/depth scale determination.

- Using LAICPMS, we can begin to assess the distribution of species within ice grains, along grain boundaries, and in veins. To date, we have looked at this list of species because they occur in relatively high abundance.

- Now, we are continuing this work to examine Na, S, and Cl to determine more about the post-depositional effects on these species. To gather a complete picture, we look at both clean and dusty ice from a range of ages.

**Specific comments by line number:**

- **Line 26-33:** Seems like this paragraph could be expanded to include more about the oldest ice and the challenges it poses, but also why it is scientifically important. In addition to work on extremely old ice, does the research presented here have any potential ability to help understand current changes in post-depositional effects on ice cores in a warming climate? If so, that would be important to mention.

- **Line 34-35:** "One very important process in this framework ... (e.g., Severi et al., 2007; Svensson et al., 2013; Fujita et al., 2015; Sigl et al., 2022)."

  It is unclear what framework we are talking about here. Post-deposition framework? In the following paragraph, there is no mention of a framework.

- **Line 45-59:** Are you looking at S, Cl, and Na because they all can (or cannot) be influenced by effective diffusivity? Why these three only? The prose is unclear, leaving the reader guessing. Clarify why your study is focused on these elements.

- **Line 49:** "For the upper 350 m in the EDC ice core drilled in central Antarctica"

  Should read "upper 350 m of the"

- **Line 85:** "Exploring the localization of impurities thoroughly over larger numbers of grains has so far been limited primarily due to methodological limitations, in particular high detection limits, hampering the collection of statistically significant large datasets that allow to draw generalized conclusions."

  This sentence is really awkward and hard to read. Consider splitting into two.

- **Line 90-101:** This entire paragraph is confusing. Are S and Cl mostly soluble like Na? How does increasing Na concentration lead to increased detection potential for S and Cl? It is not clear what you are trying to convey about the increase in Na concentration and its relationship to other elements. Is Na increasing as a post-depositional process? Or is that a depositional thing for Na to be on the boundaries and that causes S and Cl to migrate as well?

  This information needs to come earlier so that the reader is not reading for 4 pages wondering why we are looking at these specific elements.

- **Line 111-112:** We thus focus on delivering information on the localization in the ice matrix over obtaining quantitative information on concentrations, which adds considerable complexity and uncertainty (Bohleber et al., 2024).

  This concept comes out of nowhere. It would be helpful to add a sentence or two earlier in the introduction that presents this concept with more context. Why is it okay to just provide the spatial localization information? Do we need concentration data to successfully comment on the post-depositional diffusion hypotheses?

**Methods**

The methods section does not adequately explain how the study was conducted or why certain choices were made. This section needs to be revised to ensure that the methods are well explained.

First tell us about the samples you are going to work on. How you decided what to work on? Where are they from? What preparation steps did you have to go through to get the ice ready? Then, introduce the LAICPMS system(s) on which the samples were run. Tell us about how you ran them! Why did certain samples get run on one instrument vs. the other? Was there any post-processing of the data. If so, what steps did the data have to pass through?

For the LA-ICP-MS section (currently 2.1), the authors need to clarify why they ran samples at two different labs. It is unclear to the reader if the authors were attempting to have an inter-lab comparison. If so, the results and discussions are confusing because they lack a section discussing the similarities or differences observed between the two runs for the one piece of ice – sample EDC 513 – run on both machines.

Section 2.2 "sample selection" should be greatly expanded to help the reader understand where the ice is from, why specific intervals were chosen, and what preparation had to happen before the samples were inserted into the cyrostage.

**Specific comments by line number:**

- **Line 124:** "Using a NIST612 glass standard ... distance between cell and sample surface."

  Why not tune with a matrix matched ice standard?

- **Line 133:** "The latter comprises an Analyte G2 laser ablation system (Teledyne Photon Machines, USA) equipped with a HelEx II ablation chamber compatible with the Venice cryostage."

  You either need a citation for the Venice cryostage or a description here.

- **Line 137-140:** "Since using an identical ICP-TOFMS, the two systems are complementary through the differences in the laser ablation instrument."

  Sentence is unclear – incorrect word choice with "through"? Why are you using two systems? An interlab comparison? Some other reason? How did you make the decision that one lab would run large spot sizes and large areas, and that the other lab would run small, high resolution maps? If I were to attempt to replicate your method, how would I know what spatial or resolution I should map at?

- **Line 142:** "Following our previously established approach for impurity mapping (Bohleber et al., 2020), the ice surface was decontaminated by scraping with a major-element free ceramic ZrO2 blade (American Cutting Edge, USA) immediately before inserting the samples into the ablation chamber."

  This belongs in the sample selection and preparation sub-section.

- **Line 145:** What were your pre-ablation methods? How many passes did you make? Did you record the optical mosaics before or after the pre-ablaiton?

- **Line 146:** "The maps were generated using HDIP (Teledyne Photon Machines)."

  What is HDIP? Why do you need to use it to generate the maps? Is it something that is done during or following the data collection?

- **Line 151-156:** "We chose a set of ice core samples both from the Holocene and the last glacial ... to provide a realistic subsample of typical polar ice conditions."

  Can you give us more details about the cores when you introduce them here. Where exactly was it drilled. When was it drilled? Provide us with citations of other work that has been done on these cores. What time frame did you sample from? What parameters did you use to inform your sample selections? Was it purely limited by sample availability? Why did you choose these intervals of all the ice that exists? Explain not just that it is glacial, but why was *this* glacial sample selected? Are the samples from the inner part of the core? From the outer part of the core? What had to happen to the sample once you selected it? Did it have to be cut down to size? What was the cleaning procedure? Additionally, please explain why was only one sample run at both facilities? How was it determined which samples would be run where?

- **Line 153-154:** "The glacial EGRIP sample contains a dust-rich cloudy band from the Younger Dryas and thus represents contrasting conditions to the low impurity samples from EDC."

  You later refer to the samples with their sample number (i.e. EGRIP 2286). This system should be used throughout the manuscript for clarity.

- **Line 155:** 250 000 particles/ml

  Typo: 250,000 particles/mL

**Results**

As currently written, the results and discussion sections are difficult to follow. Since The Cryosphere has no strict section requirements, consider if blending the results and discussion sections into a single section would help readability. Framing the section as:

> To examine XXXX we made this map on this sample. We find XXXX. To better understand XXXX, we analyzed the sample again higher resolution (?). To examine this other hypothesis, we made a map of this sample XXXX, and found XXXX.

This would allow you to form a cohesive narrative that would benefit the reader's comprehension of the science.

**Specific comments by line number:**

- **Line 162-163:** "Figures 1 and 2 show examples for the large size $^{32}$S intensity maps alongside $^{37}$Cl and $^{23}$Na recorded with the Graz LA-ICP-TOFMS system."

  This is the first time you have presented isotopes of elements. Since you will later go on to show maps of multiple isotopes for each element, it would be good to add some text in the methods explaining what isotopes you gathered and why it is helpful to look at more than one.

- **Line 163:** "Additional maps are included in the Supplementary Material"

  Which maps are you referring to here? All of the supplementary material. There are no other references to Figure S1 and Figure S3. Are they needed? If they are adding something important, they should be discussed in further detail somewhere in the manuscript.

- **Line 163-165:** "Na serves as a reference element showing the high degree of localization at grain boundaries, analogous to previous observations in EDC and EGRIP (Bohleber et al., 2020; Bohleber et al., 2023; Stoll et al., 2023).

  This is a super important sentence and needs to be clearly stated in the introduction.

- **Line 167-170:** "The EGRIP cloudy band sample likely has significantly higher impurity content compared to the EDC Holocene sample, reflected in the higher signal amplitude in spite of the smaller spot size used (20 µm EGRIP vs 40 µm EDC)."

  What is the sample number of the EDC sample? Why were the samples run at different spot sizes? This needs to be explained with more clarity because it is not clear that comparing different spot sizes is a good idea. Are you comparing apples and oranges, or is it okay to compare like this?

- **Line 183-187:** "Using the AWI setup, maps were recorded on two subsamples ... the map on EDC513-1 (shown in the Supplementary Material)."

  This should not be the first time that we are learning that EDC 1819 and EDC 513 were subsampled. This information should be provided in the "Sample Selection" portion of the methods. It should also be included in Table 2. It is unclear why it matters what order the samples were mapped in. If it is important, then the order of mapping should be included in the methods section. It would be helpful if you expanded the discussion of Figures 3-5. A brief description of each figure, including the sample, pixel size, and notable features, is warranted. What supplemental figure is EDC 513-1?

- **Line 189-190:** "The map for $^{35}$Cl contains only noise in all maps recorded (hence not shown), but strong signal in $^{37}$Cl, which indicates the potential formation of Chlor- hydrogen adducts (see Sect. 4.1)."

  It may be helpful to show both isotopes as a figure in the supplement. That way the reader doesn't have to trust you at your word. The discussion about Chlor-hydrogen adducts would be helpful to have right after you introduce this so the reader can follow easier.

- **Line 190:** "As shown in Fig. 3 – 5 for S"

  If you are not going to discuss each figure in detail, it is strange to have all three. Why show them if they are not noteworthy?

**Discussion**

**Specific comments by line number:**

- **Line 223-225:** "Although Na, S and Cl belong to the most abundant elements in polar ice (albeit generally at low ppb bulk concentrations)(Legrand & Mayewski, 1997), initial studies in LA-ICP-MS ice core analysis and in particular impurity mapping have focused on analytically more easily accessible elements such as Na, Mg, Al, Ca, Fe, Sr"

  This idea is so crucial to the understanding of your research. This needs to be said *way* earlier, like in the introduction!! The text would also benefit from an additional prose about why Na, Mg, Al, Ca, Fe, and Sr are more accessible.

- **Line 226:** "As demonstrated by the results shown in Fig. 1 – 7"

  There are only six figures in the main text. Where is figure 7?

- **Line 228-260:** "The main hurdle for detecting S is the mass interference by ... collision-reaction cell gases, such as Xe or $CH_4$ (Guillong et al., 2008; Singh et al., 2025)."

  Both of these paragraphs need to be moved into the methods section when you talk about LA-ICP-TOFMS setup and analysis methodology.

- **Line 290:** "the higher amount of resources consumed"

  What kinds of resources? Ice? Or analytical resources (time, energy, gases)? You make a reference of 200k shots for the map in Figure 6, but the reader has no idea how many shots were needed for other maps. Therefore, it is hard to understand the increase in resources.

- **Line 291:** "Consequently, we only investigated this exemplarily in this work."

  There appears to be some missing text in this sentence.

- **Line 294-295:** "While at 10 µm spot size, grain boundaries are typically mapped with 2-3 high intensity pixels, for 5 µm this reduces to 1-2 pixels, with the same dosage 10 for all maps."

  It is unclear how the number of high intensity pixels translates into data quality.

- **Line 298-299:** "The 1 µm spot size may be more capable of resolving the fine-scale impurity variability around triple junctions, and shows that some enhancement at triple junctions relative to grain boundaries may exist at that spatial scale"

  Enhancement of what? High intensity pixels?

- **Line 299-300:** "In previous studies a comparable spatial resolution around 1x1 µm was used."

  Citations needed.

- **Line 324-327:** "Accordingly, the physical reality likely is a mix between the two scenarios described in Ng (2021), with a highly localized nature of impurities in ice but variable contributions from ice veins, grain boundaries and grain interiors depending on conditions, such as climate period and insoluble particle content, and uncharacterized microstructural processes."

  This sentence is hard to understand. It would benefit from being split up into multiple sentences and more detail added.

- **Line 329-331:** "Notably, the situation in deep ice sections, with larger grains and more time spent at higher temperatures, may deviate from the snapshots from relatively shallow depths presented here and remains to be explored, in particular regarding the influence of geochemical reactions leading to dissolution at grain boundaries (Baccolo et al., 2021)."

  Why has this research been done at relatively shallow depth intervals, if developed methodology is supposed to be used on ultra-deep ice? Is it an ice availibility problem? How do you know that your method will still work on deeper ice core samples?

- **Line 331-332:** "Technically, the same is true for sections with high S-peaks from volcanic eruptions, which have not been measured here."

  Please add citations to papers where the high S peaks have been measured, so the reader can learn more and see how your data compares.

**References**

Many of the citations are incomplete, lacking dois and other information. In addition, your citations do not follow the format required by the journal. A complete list of reference requirements can be found at https://www.the-cryosphere.net/submission.html. Note that the year is at the end of the citation and that no part of the citation includes italicized text. Please work through the citations, fixing formatting and adding information where possible.

- **Here is the style for journal articles**: Porter, J. G., De Bruyn, W., and Saltzman, E. S.: Eddy flux measurements of sulfur dioxide deposition to the sea surface, Atmos. Chem. Phys., 18, 15291–15305, https://doi.org/10.5194/acp-18-15291-2018, 2018.

- **And the style for books**: Singh, O. N. and Fabian, P. (Eds.): Atmospheric Ozone: a Millennium Issue, Copernicus Publications, Katlenburg-Lindau, Germany, 147 pp., ISBN 393658608X, 2003.

**Tables**

**Table 1:**

Consider reformatting this table so that it is easier to compare the laboratories with each other. This could be achieved in three columns: setting, value for AWI, and value for Graza. Why are the TOF settings not the same in the two labs? Is it because of the different lasers? What dictated changes to the rep rate?

The authors should have an expanded version of this table, either here or in the supplement, that lists all of the settings that were used. Note that these lists were generated for sample analysis on a Thermo Element 2, Thermo Element XR, and an Agilent 8900. Potentially, some settings are listed that are not appropriate for ICP-TOFMS or settings that need to be added to the list for ICP-TOFMS.

**For the laser ablation system:**

- Instrument Make/Model
- Software
- Ablation Cell
- Laser Wavelength
- Pulse Width
- RF Power
- RF Matching

- Sample Depth
- Gas Flow Setup (i.e. Direct connection)
- Carrier Gas Flow
- Energy Density
- Repetition Rate
- Spot Size
- Sampling Pattern (i.e. Single hole drilling or raster)

- Ablation Duration

**For the ICP-TOFMS system:**

- Instrument Make/Model

- Software

- Sample numbers measured

- RF Power

- Plasma Gas Flow

- Auxiliary Gas Flow

- Nebulizer Gas Flow

- Makeup Gas Flow

- Reaction Gas

- Reaction Gas Flow

- Analyzer Pressure

- Sample Inlet

- Signal Smoother

- ARIS rapid aerosol introduction system settings

- Nebulizer make/model

- Sample uptake rate

- Sample Cone make/model

- Skimmer Cone make/model

- Torch Z position

- Sensitivity

- Scan Type

- Mass Window

- Integration Time or Window

- Detection Mode

- Isotopes Measured

**Table 2:**

This table should be expanded to include the laser spot size(s) used to help the reader understand what ice was measured with what settings. This is especially important because this information cannot easily be gleaned from Table 1. For audiences with broad backgrounds, consider adding a column noting if the sample is Holocene or Last Glacial.

**Figures**

**General comments:**

Consider moving the element labels to the left y axis. This would allow you to stack the maps close together for ease of visual comparison. The scale bars would benefit from a white box behind it or to be moved off of the map. It is very difficult to read. Consider adding a scale bar to each image/map if you choose to leave them as is (not closely stacked). Does cts = counts per second (cps)? Cts should be explained. Consider adding more details to the figure caption. Talk about the sample time frame that we are looking at and any notable features that the reader should focus on. Please reference the specific figures in the supplement instead of just saying "Supplementary Material".

**Specific Comments:**

**Figure 2:** Make sure that your optical mosaic is cropped to the same extent as the LA-ICP-TOFMS elemental maps.

**Figure 6:** It seems that this figure would work better as a 3 panel strip; in each row – optical, 10um, 5um, 1um for a different element/isotope. If you keep it as is, please add an element/isotope label to the 5 um and 10 um spot size panels.

**Figure S2:** Is it possible to adjust the contrast on the mosaic image to improve the visibility of the grain boundaries?

**Figure S3:** The intensity is now in au. Why the shift from cts? What is au?

---

## Author Comment (AC1)

**Response to referee #2**

The paper by Bohleber et al. presents the first attempt at two-dimensional mapping of S and Cl in EDC and EGRIP deep ice cores using LA-ICP-MS. Although both elements are relatively abundant in these cores, due to analytical challenges with ICP-MS, they have not been previously targeted by LA-ICP-MS studies. This paper shows that signals of S and Cl can be detected even in samples that do not contain elevated concentrations of volcanic S and Cl. Since both S and Cl are important for understanding mechanisms and extents of impurity diffusion in ice—crucial for interpreting the chemical signatures of "Oldest Ice"—the new methods developed in this study have great potential for future ice core research. The paper also demonstrates that not only the most abundant isotope of S, but also less abundant isotopes can be detected. This would contribute to investigating sources of S. Therefore, I strongly support the publication of the fascinating new results presented in this paper.

We thank the reviewer for the appreciation of this work and the constructive comments, that helped us to improve and expand in particular the presentation of the results and the discussion part of the manuscript.

However, I have a question or concern regarding the conclusion about the localization of S. The authors find a high level of localization of S at grain boundaries, with only minor occurrences within grain interiors even in glacial ice. Previous studies using Raman spectroscopy (Ohno et al., 2006; Sakurai et al., 2011; Stoll et al., 2021) reported numerous S-containing salts or minerals such as $Na_2SO_4$ and $CaSO_4$. I wonder why the samples used in this study did not contain many of these S-containing particles in grain interiors. Although there may be differences between Dome Fuji (Ohno et al., 2006; Sakurai et al., 2011) and EDC samples (this study), Stoll et al. (2021) used EGRIP samples, as in this study. I wonder if this discrepancy is due to a sample-to-sample (or layer-to-layer) difference, or if it might be an artifact of the analytical methods used here. I would like to see some explanations about the difference between the findings of this study and those of previous studies.

This is a very good point which we have now addressed in more detail in the discussion. First and foremost, we note that the EGRIP sample does show some isolated clusters of high intensity pixels in grain interior locations for Na and S – some co-localised and some with only S. Also in EDC, e.g in sample EDC1819-2, we find high intensity spots in the grain interior. We have highlighted this better in the Figures, and also stress this finding more in the discussion.

In general, we find that comparing the data generated by two different methods is extremely difficult. In two earlier papers (Stoll et al., 2023; Bohleber et al., 2023) we crossed this inter-method bridge between Raman spectroscopy and LA-ICP-MS by analysing the same very dust-rich samples of Greenland cloudy bands with both methods resulting in comparable results. Here, the dust is so abundant and often clustered (larger size) which greatly aids its detection in LA-ICP-MS mapping. Some results showed promising consistencies, e.g. the relative fraction of particles at grain boundaries being very similar seen in both techniques (Bohleber et al., 2023).

We do not believe that differences are related to artefacts. It should be emphasized that LA-ICP-MS and single particle Raman spectroscopy have very different analytical figures of merit which provide insights into different aspects of insoluble particles and soluble species. Raman spectroscopy is applicable to look at a limited number of large particles (diameter of 1 micrometer or above) but cannot resolve smaller particles or ionic (dissolved) fractions

(see Stoll et al. 2021, 2022, 2023). LA-ICP-MS detects both fractions but does not allow a differentiation between ionic and particulate element species. We have spotted several instances of isolated hotspots within the crystal structure, which we suspect to be associated with discrete insoluble particles. However, grain boundaries are composed in a more complex fashion and a definite differentiation was not possible. As such, Raman spectroscopy and LA-ICP-MS are not expected to directly validate each other but rather to grant complementary insights.
Notably, the same situation applies to the comparison with the pioneering works investigating S in ice with SEM-EDS (e.g. Mulvaney et al., 1988). Although in this case we have consistent findings with LA-ICP-MS regarding the presence of S at triple junctions. As explained in the text we suspect that limits of detection may not have been sufficient to detect S at grain boundaries with SEM-EDS.

The paper states that the new impurity maps support a view of diffusive transport not only through ice veins but also along grain boundaries, yet do not show clear differences between samples from the Holocene and the last glacial period in the EDC core. Based on these findings, the authors seem to reject the hypothesis by Rhode et al. (2024) that the diffusion mechanism changed between the Holocene and the last glacial period due to changes in localization. However, if the amount of S in grain interiors changed between the Holocene and the last glacial period but was not detected by the methods used in this study or not observed in the specific samples analyzed by chance, the authors cannot completely deny the hypothesis. Therefore, I would like the authors to confirm that the very low amount of S detected in grain interiors in glacial EDC ice is robust and not an artifact caused by analytical issues.

We can confirm that the low amount of S in the grain interiors is robust and consistent with other data generated on EDC samples before and after this study. To be fully clear: Small amounts of S may be present in the interior (see comment above). Regarding the implications for the hypothesis by Rhodes et al. (2024), we have rephrased the respective text in the discussion to make clear we are not completely discarding this hypothesis based on our, admittedly, limited "snapshots". We would also like to point out that our data is consistent with a recently formulated view that the majority of the diffusion is triggered at shallow depths and that only minor differences occur after (Ng et al., 2025).

Ng, F. S. L., Rhodes, R. H., Fudge, T. J., and Wolff, E. W.: Doomed descent? How fast sulphate signals diffuse in the EPICA Dome C ice column, EGUsphere [preprint], https://doi.org/10.5194/egusphere-2025-1566, 2025.

Additionally, I have some minor editorial comments, which are listed in the "Detailed comments" below.

**Detailed comments:**
1. **Section 2.1** – This section needs more details on how this study succeeded in detecting S and Cl using LA-ICP-TOFMS. I recommend moving a major part of Lines 223–251 to Section 2.1, as this part is more appropriate in the Methods section rather than the Discussion.
   Changed accordingly, also to address comments by reviewer #1.

2. **Line 145, pre-ablation** – Please explain more about pre-ablation. If this involves sublimation of the sample surface, I have a concern that grain boundaries might be preferentially sublimated compared to grain interiors, which could lead to concentrated impurities at grain boundaries. Please confirm that this is not the case.

This is an important issue which we have taken seriously from the beginning starting with LA-ICP-MS mapping on ice. So far, we have concluded that the grain boundary localization is not an artifact caused by sublimation, although we do not feel comfortable with a giving an ultimate answer (if there is ever such a thing in science). Notably, already in the first study (Bohleber et al., 2020) we experimented with various consecutive maps taken on top of each other, each exposing a new surface- and including further experiments with preparing fresh surfaces for sequential mapping of the same area. The purpose of pre-ablation is to remove the surface layer immediately before analysis, providing an additional decontamination step. How thick this layer is, is hard to estimate, but visible ablation is traceable in general, especially with larger spots.

We have also investigated the effects of repeated ablation on the same ice core area to investigate to what extent elemental maps can be obtained repeatably. These experiments have shown qualitatively the same elemental distributions, which is a clear indication that sublimation and the artefact generation as suggested is not a driving mechanism.

We further would like to emphasize again that intra-grain intensities do occur, so the grain boundary localization is not the full story. Moreover, not all impurities are found at grain boundaries by LA-ICP-MS mapping. Impurities with a significant insoluble fraction like Al, Si and Fe can also lack the grain boundary association (e.g. for the cloudy band sample in Bohleber et al., 2023), or show it much less pronounced. Recent calibration experiments provided an additional route for cross-checking the LA-ICP-MS maps: we found that the concentrated impurity values at the grain boundaries are consistent with (lower) bulk concentration levels measured on the same sample after melting. We further found that the comparison with meltwater analysis revealed that the LA-ICP-MS maps correctly predicted which samples had a substantial contribution from insoluble particles (Bohleber et al., 2024).

In summary, we have found no evidence for the grain boundary localization being an artefact. Especially no evidence of a time-dependent effect changing the impurity distribution on the ice surface was found, which would arguably be the case if sublimation was the dominant cause of the grain boundary signal.

3. **Table 2** – Please add concentrations of S (or $SO_4^{2-}$ would be fine), Cl, and Na for the samples used in this study. If the data from exactly the same depths are unavailable, concentrations from similar depths would be helpful to better understand the differences among samples.

For the exact depths we unfortunately do not have any data. We intend to keep the samples used for this study available for further analysis, thus did not melt them for analysing liquid concentrations (since not in focus here). The EGRIP CFA chemical record is work in progress and, so far, no data on S or SO4 concentrations is available.

4. **Figures 1, 2, 4, and S2** – It is difficult to see grain boundaries and air bubbles in the optical mosaic images. Please add optical mosaic images with marked grain boundaries and air bubbles for each figure. Figure S1 also needs an optical mosaic image to show grain boundaries.

Changed accordingly.

5. **Lines 166–167, "In EGRIP 2286…… (Fig. 2)"** – It is hard to identify the isolated pixels within the grain interior showing high S values. Please mark these pixels clearly.
Changed accordingly. Note these pixels in relation to the first comment above about detecting mineral dust (and salt) particles.

6. **Lines 167–169, "The EGRIP cloudy……… (40 μm EDC)"** – It is hard to see the intensity difference between EGRIP and EDC. To confirm this, the authors need to show the intensity data.
We are not sure we fully understand this request, as the intensity is already shown as the scale on the color map of figures 1 and 2, and the color-coded maps provide arguably the best representation of our 2D data. However, considering carefully this comment and the following 7. & 8., we made an attempt for an alternative quantitative overview, trying to address the comments 6,7 and 8 here.

7. **Lines 169–170, "Both maps show………… a few mm"** – Please show the intensity data for the grain boundaries.
We would like to point out that with this part of the text we referred to the spatial variability in the grain boundary intensities – something you can only show with the 2D maps. We tried to illustrate that with a concrete example in lines 171-173.
In the context of comments 6, 7, 8: We understand this as a request to separate out the intensities at the grain boundaries from our data, and have come up with the following solution: First, to show the entire dataset we calculated the histogram of the intensity values of the entire map (shown exemplarily in the Figure below – to be included in the manuscript or the Supplementary Material). Here we typically observe a bimodal distribution with one mode corresponding to the low intensity grain interiors and another mode matching the high intensities at grain boundaries. To confirm this, we segmented out the pixels belonging to the grain boundaries using a watershed algorithm (as in Bohleber et al., 2023). The red dot and line indicate the median and its interquartile range of the grain boundary pixels, respectively.

[Figure]

Figure xy: Intensity distributions shown as histograms for Na & S of Figure 1 and 2 in the main manuscript. Note the bimodal distribution, especially visible for Na, which correspond to high intensity foreground (e.g. grain boundaries) and low intensity background (grain interiors) of the image. The inserts in the Na histograms show the grain boundary segmentation. Red dots denote the median of the thereby segmented grain boundary pixels, the red line extends within the 25-75% interquartile range. Yellow triangles denote the median intensity of pixels at the triple junctions (segmented manually).

8. **Lines 173–174, "At 40 µm and ……… at triple junctions"** – I'm not convinced by this sentence. Some triple junctions appear to show strong intensities. The authors need to provide more quantitative discussion here, as I wrote in comments 6 and 7. It is indeed hard to assess intensity variability just from visual images.
   Our main point here was that we believe this to be a (partial) matter of resolution, as only very high resolution (1 µm) may be able to resolve the concentration differences between the triple junction and its adjacent grain boundary due to the very small size of theses microstructural features. To further substantiate the sentence referred to here we now include an exemplary investigation into the triple junctions of Figures 1 and 2. We manually segmented the pixels belonging to triple junctions in both maps, and calculated the respective median value, which is shown as a yellow triangle in the Figure above. It becomes clear that, albeit slightly elevated with respect to the median of the grain boundaries, the triple junction value in all cases still falls within the interquartile range. We thus regard this as no clear evidence for a clear

enhancement at the triple junction – but again, we stress that this is likely connected to the spatial resolution of 40 and 20 µm not being sufficiently high.

[Figure]

[Figure]

Figure xy: Manual segmentation of the pixels belonging to triple junctions in Figure 1 and 2, shown on top and bottom, respectively. Red crosses show the segmented pixels at triple junction. Their median values are shown in the Figure above as yellow triangles.

9. **Figures 3 and 4** – Please add the spot size in the figure captions. Although it is mentioned in the text, including it in the captions would make it easier to follow.
Changed accordingly.

10. **Figures 1, 3, 4, 5, 6, S1, S2, S3** – Please use larger fonts in the diagrams.
Changed accordingly.

11. **Line 226, "Fig. 1–7"** – This should be "Fig. 1–6." There is no Fig. 7.
Changed accordingly.

12. **Line 250, "It is likely that also ³⁷Cl is mass-shifted and detected at ³⁹K instead."** – If this is the case, please explain why the authors can still distinguish between K and Cl signals. Why can we be sure that the signals in the maps represent Cl?
We can clearly distinguish Cl and K by taking the isotopic abundance of Cl into account. The mass shifting of Cl is an element-specific feature, which happens to Cl alone. As such, $^{35}Cl$ and $^{37}Cl$ are mass shifted to $^{35}Cl^1H_2^+$ and $^{37}Cl^1H_2^+$, respectively. The latter has a spectral overlap with K and on its own, it cannot be differentiated from $^{39}K$. However, the former ($^{35}Cl^1H_2^+$ ) does not show any overlap with other elements and can selectively be detected as proxy for the Cl distribution.

13. **Lines 286–288** – Why do the intensities at grain boundaries depend on scan direction? Why does this dependence only appear for the 1 µm spot size? Unless a clear explanation is given, I'm concerned about the authors' discussion on differences in signal intensities.
We do not have an explanation for this effect yet as LAICPMS analyses at 1 µm are novel to ice core research and we are investigating it further. While refraining from

speculation we still wanted to point it out, however, for full transparency. We did not observe this dependency on scan direction at coarser spot sizes (e.g. 10 μm), and thus the vast majority of the data. Investigating this issue further is beyond the scope of this study and could be the main topic of discussion in a future study.

14. **Lines 313–322** – Considering previous studies, I would expect higher concentrations of Na in grain interiors. However, this is hard to see in Figures 2, 3, or 4. Please mark Na in grain interiors more clearly.
Changed accordingly. We also note again, that the EGRIP map (Figure 2) is consistent with what we have previously observed in cloudy band samples from Greenland (Bohleber et al., 2023, Stoll et al., 2023).

15. **Lines 332–334, "At least for sections… bulk S concentration."** – I agree that it is difficult to see systematic differences between the data presented here. However, if the ratio of S at grain boundaries to grain interiors changes (but not detected by the methods used in this study), it could affect the apparent diffusivity. To draw a conclusion, I think more quantitative analysis is needed.
To be clear: We can observe intra-grain concentrations – the primary challenge is not the localization but the detectability. If individual minerals / salts are much smaller than the spot size, the resulting intensity contrast becomes weaker and harder to detect. This means that, even if present, the relative fraction of the grain interior contributions to the bulk concentration is small for Na, Cl and S – hence also their expected contribution to alterations via diffusion. Within these limitations, there is no systematic difference in the impurity localization, providing valuable constraint for discussion potential impurity diffusion mechanisms.

16. **Lines 346–349** – From Figures 3 and 4, it is difficult to see partially interrupted impurity populations at grain boundaries and air bubbles in glacial maps. Please show these features more clearly.
Changed accordingly.

17. **Lines 357–365** – If S/Na and Cl/Na at grain boundaries are similar for Holocene and glacial samples, I don't think this necessarily argues against a relative difference. Changes in the amounts of these elements in grain interiors—possibly not observed by the methods used—could result in apparently different diffusivity.
See our reply to comment 15. One of the primary merits of the maps shown here is that they can constrain the contributions by grain boundaries and interiors. From the limited snapshots that we can provide so far, in-grain concentrations should not be drastically different, or they would have been rising to detectable limits. We would also like to stress that, in the previous direct comparison with liquid ICP-MS analysis (Bohleber et al., 2024) we found no evidence for a systematic underestimation of concentrations by LA-ICP-MS, which would indicate that we are systematically missing some fraction due to limits of detection. This is not the case.

18. **Lines 384–385, "but do not show any …. EDC ice core."** – If impurity localization at grain boundaries and veins shows no clear differences, differences in grain size would change the ratio of grain boundaries and veins in a unit volume, potentially affecting diffusivity. However, smaller grain sizes in glacial ice would give larger ratios of grain boundaries and veins, leading to faster diffusion, correct? I think grain size data are also important for considering diffusion mechanisms.
We agree that grain size variability and/or rate of grain growth *may* be important determinants on diffusion rate, depending on the mechanism(s) acting. Barnes et al., 2003 provide two different sulfate diffusion mechanisms that both imply increased

diffusivity with increased grain growth rate. However, this relationship was not observed for EDC sulfate by Rhodes et al., 2024 or Fudge et al., 2024. But, it would be most recognizable in deeper EDC ice than investigated by those studies (or here) where temperatures exceed -10C and migration recrystallisation (rather than normal grain growth) occurs.

Ng's 2021 proposed diffusion mechanism within veins is 'independent of grain growth and occurs in the absence of grain-size variations'. Ng (2021) also defines 'residual diffusion' resulting from vein motion due grain boundary migration (normal grain growth assumed). Ng concludes that diffusivity is independent of grain diameter even in this case because "smaller grains lead to faster grain boundary migration but proportionally shorter mean free path for the vein motion" (see his Appendix B). Smaller grain sizes and therefore greater area of veins and/or grain boundaries per unit volume will not automatically cause faster diffusion. Diffusivity is not reported per unit volume (unit: m2yr-1). The level of interconnectedness between veins and grain boundaries, which may be impeded by bubbles or particles, will also play a role.

**References**

- Barnes, P. R. F., Wolff, E. W., Mader, H. M., Udisti, R., Castellano, E., & Röthlisberger, R. (2003). Evolution of chemical peak shapes in the Dome C, Antarctica, ice core. Journal of Geophysical Research: Atmospheres, 108(D3).
- Bohleber, P., Larkman, P., Stoll, N., Clases, D., Gonzalez de Vega, R., Šala, M., ... & Barbante, C. (2024). Quantitative insights on impurities in ice cores at the micro-scale from calibrated LA-ICP-MS imaging. Geochemistry, Geophysics, Geosystems, 25(4), e2023GC011425.
- Bohleber, P., Stoll, N., Rittner, M., Roman, M., Weikusat, I., & Barbante, C. (2023). Geochemical Characterization of Insoluble Particle Clusters in Ice Cores Using Two-dimensional Impurity Imaging. Geochemistry, Geophysics, Geosystems, 24(2), e2022GC010595.
- Bohleber, P., Roman, M., Šala, M., & Barbante, C. (2020). Imaging the impurity distribution in glacier ice cores with LA-ICP-MS. Journal of Analytical Atomic Spectrometry, 35(10), 2204–2212.
- Fudge, T. J., Sauvage, R., Vu, L., Hills, B. H., Severi, M., & Waddington, E. D. (2024). Effective diffusivity of sulfuric acid in Antarctic ice cores. Climate of the Past, 20(2), 297-312.
- Mulvaney R, Wolff EW and Oates K (1988) Sulphuric acid at grain boundaries in Antarctic ice. Nature, 331(6153), 247–249 (doi: 10.1038/331247a0)
- Ng, F. S. (2021). Pervasive diffusion of climate signals recorded in ice-vein ionic impurities. The Cryosphere, 15(4), 1787-1810.
- Rhodes, R. H., Bollet-Quivogne, Y., Barnes, P., Severi, M., & Wolff, E. W. (2024). New estimates of sulfate diffusion rates in the EPICA Dome C ice core. Climate of the Past, 20(9), 2031-2043.
- Stoll, Nicolas, Eichler, J., Hörhold, M., Shigeyama, W., & Weikusat, I. (2021). A review of the microstructural location of impurities in polar ice and their impacts on deformation. Frontiers in Earth Science, 8, 658
- Stoll, N., Hörhold, M., Erhardt, T., Eichler, J., Jensen, C., & Weikusat, I. (2022). Microstructure, micro-inclusions, and mineralogy along the EGRIP (East Greenland Ice Core Project) ice core–Part 2: Implications for palaeo-mineralogy. The Cryosphere, 16(2), 667-688.
- Stoll, N., Westhoff, J., Bohleber, P., Svensson, A., Dahl-Jensen, D., Barbante, C., and Weikusat, I. (2023). Chemical and visual characterisation of EGRIP glacial ice and cloudy bands within, The Cryosphere, 17, 2021–2043, https://doi.org/10.5194/tc-17-2021-2023.

---

## Author Comment (AC2)

**Overall quality of the preprint manuscript**
This preprint manuscript presents new findings on the spatial distribution of sulfur (S), chlorine (Cl), and sodium (Na) in polar ice using high-resolution laser ablation inductively coupled plasma mass spectrometry (LA-ICP-TOFMS). The study focuses on samples from the EPICA Dome C (EDC) core in Antarctica and the EGRIP core in Greenland, aiming to understand impurity localization and its impact on post-depositional diffusion. The results show that S, Cl, and Na are predominantly localized along grain boundaries, with limited evidence for accumulation at triple junctions or within grain interiors, except in dust-rich ice. These findings suggest that diffusion may occur primarily along grain boundaries rather than through interconnected veins, challenging assumptions that post-depositional mobility varies significantly with climatic periods. The study highlights the need to integrate impurity mapping with modeling efforts to better constrain diffusion processes that affect paleoclimate records preserved in deep ice cores.

While the fundamental science discussed in the preprint is a significant contribution to the field and is clearly within the scope of The Cryosphere, the manuscript text requires substantial reworking before it is suitable for publication. In particular, the framework established in the introduction and methods sections lacks clarity and cohesion, making the central aims and motivations of the study difficult to follow. This undermines the accessibility of the results and their implications, even for readers familiar with ice core and/or LA-ICP-MS science. Once the manuscript has undergone major revision, it will be a lovely addition to The Cryosphere.

We thank the referee for her effort in providing help in making the manuscript more accessible to the readers of TC. We generally followed these suggestions and believe it has significantly increased the clarity of the text.

**Individual scientific questions/issues/comments**
For ease of reading, specific comments, questions, and issues have been broken into sections. Generally for each section, a list of small, specific comments by line number follows the more general comments.

**Abstract**
The abstract does a great job of summarizing the work presented. There are just a few comments to help readers as they are introduced to the presented work.

**Specific comments by line number:**

• Line 12: "Na, S and Cl are among the relatively abundant impurity ... precise physical mechanisms remain unclear."
Does one expect that similar post depositional effects are seen in Na and Cl?
Very generally, each chemical impurity could be affected differently by post-depositional movement. Differences may arise based on ionic radius and charge (cations and anions) as well as localization within the ice. This is a highly simplified answer, of course. The second sentence quoted here refers to diffusion, which may affect not exclusively S (or sulfate) – and for this the physical mechanism remains to be fully identified. This is where we hope to make a contribution with the manuscript – complementary to modelling and data-based

efforts investigating peak-broadening. Due to the brevity of the abstract, not much has been added but this aspect is picked up on in the revised introduction.

Do we know enough about the system to know that there is an element that does not suffer post-depositional changes?
The short answer: No. The application of LA-ICP-MS is an entirely new paradigm to study the distribution of elements at the microscale and it ultimately might contribute to an enhanced understanding how elements behave during post-depositional changes. However, we are only at the beginning and further investigations are required.

• Line 15: "Mapping the two-dimensional impurity distribution ... S and Cl have not been targeted thus far."
Can you give us an example of what elements work well?
S and Cl are traditionally elements which are difficult to analyse in ICP-MS due to spectral interferences. The here developed methods provides new options to study these elements in ice and allows correlative studies to other elements. Typically, metals are significantly easier to analyse in ICP-MS and consequently, elements such as Na, Mg, Sr can be resolved easily. We have added this to the manuscript as "e.g. for Na, Mg, Sr, …",

• Line 17: "We show here that signals of S and Cl can be detected in ... exemplarily even 1μm."
The grammar at the end of the sentence ("exemplarily even 1μm.") is really awkward. Please rewrite for clarity.
Changed accordingly.

• Line 19-20: "We find a high level of localization of S and Cl (and Na) at grain boundaries but also some dispersed occurrence within grain interiors in dust-rich ice."
Why is Na pulled out into parentheses? Is this a paper about S, Cl and Na? Or about S and Cl?
The paper is primarily about Cl and S, being the new analytes. Na provides an important reference, e.g. to previous LA-ICP-MS maps. Hence the parentheses. After more consideration, we will remove Na from the title of the manuscript to make it clearer that the novelty of the paper comes through the investigation of Cl and S, while Na serves as a reference element.

• Line 20-23: "The new maps support a view on diffusive transport not only through ice veins but also along grain boundaries, but do not show any clear differences in this regard between samples from the Holocene and last glacial period in the EDC ice core."
This sentence is long and bulky. Consider splitting it into two. Also you have not defined EDC yet, please do so here. The casual reader, even someone who works in the field will likely not know what you are talking about.
Changed accordingly.

**Introduction**
While the introduction does introduce concepts relevant to the manuscript and science, the structure of the introduction is disjointed and the message muddled. The majority of the introduction is dedicated to understanding a single element, S. Small interjections mention Cl and Na as well, but the reader is left confused as to why we are looking at three elements,

if only one is of importance. There are also large leaps in text, rather than presenting a cohesive narrative.

It seems best to take a step back and outline the key information that needs to be conveyed. Once a clear flow of information has been outlined, the narrative will quickly come together. Most of the pieces are already there. Perhaps something like this?

• First, talk about the fact that we work with a fundamental assumption that there is no post-depositional migration of species. However, we know that this assumption is a simplification.
Here are some post-depositional effects that we are really concerned about. Critically, ice cores that have much to tell us about the very distant past are at the highest risk of experiencing these migrations. Specifically, the community knows that species X¡ Y¡ Z are of particular concern. There is a high level of concern about S because we know that it migrates at depth and is one of the underpinning species for age/depth scale determination.

• Using LAICPMS, we can begin to assess the distribution of species within ice grains, along grain boundaries, and in veins. To date, we have looked at this list of species because they occur in relatively high abundance.

• Now, we are continuing this work to examine Na, S, and Cl to determine more about the post-depositional effects on these species. To gather a complete picture, we look at both clean and dusty ice from a range of ages.

We are grateful for the comments (including the ones to follow) that helped to restructure the introduction. The point is that while the original study by Barnes et al. (2003) also considered Na, Cl, and S – in most of the following studies the study of diffusion is connected to S/sulfate. In the introduction we need to discuss the state of knowledge regarding the diffusion mechanism and where it is coming from (modelling vs data, analytical techniques). Thus, a natural focus on S/sulfate develops in the introduction. However, we emphasize now that, in this work, all three elements are important. We have tried to integrate better Na and Cl, while also emphasizing the role of Na as a reference element.

**Specific comments by line number:**
• Line 26-33: Seems like this paragraph could be expanded to include more about the oldest ice and the challenges it poses, but also why it is scientifically important. In addition to work on extremely old ice, does the research presented here have any potential ability to help understand current changes in post-depositional effects on ice cores in a warming climate? If so, that would be important to mention.
We generally tried to keep the introduction as concise as possible but are now including a reference to the "Mid-Pleistocene Transition", which is driving the current Oldest Ice efforts (Fischer et al., 2013, Wolff et al, 2022). The suggestion about a connection to ice in a warming climate is very interesting, but this would likely concern mostly the uppermost layers in the ice sheet – if not being as dramatic as leading to temperate conditions and impurity elusion, but this would be a different mechanism requiring a separate study.

• Line 34-35: "One very important process in this framework ... (e.g., Severi et al., 2007; Svensson et al., 2013; Fujita et al., 2015; Sigl et al., 2022)."
It is unclear what framework we are talking about here. Post-deposition framework? In the

following paragraph, there is no mention of a framework.
We removed the work "framework" and clarified this paragraph.

• Line 45-59: Are you looking at S, Cl, and Na because they all can (or cannot) be influenced by effective diffusivity? Why these three only? The prose is unclear, leaving the reader guessing. Clarify why your study is focused on these elements.
S and Cl are difficult to resolve in LA-ICP-MS and with the here presented method, it becomes possible to study their distribution at micrometer resolution. The focus on S and Cl is related to their relevance in post-depositional changes. However, Na can be seen as a reference element, which can be found more easily, and which allows comparison of the distribution of the respective elements. The extent to which these elements are affected by post-depositional processes may vary depending on species and process – e.g. this is not just connected to diffusion, but also other processes, such as salt-formation and displacement by grain growth. Another reason is related to the original study by Barnes et al. (2003) providing an important reference for Na, Cl and S. Furthermore, we focus on only three (not 30) elements to enable a concise analysis (due to the analytical limitations and challenges mentioned below) and interpretation. We clarified this motivation in the text.

• Line 49: "For the upper 350 m in the EDC ice core drilled in central Antarctica"
Should read "upper 350 m of the"
Changed accordingly.

• Line 85: "Exploring the localization of impurities thoroughly over larger numbers of grains has so far been limited primarily due to methodological limitations, in particular high detection limits, hampering the collection of statistically significant large datasets that allow to draw generalized conclusions."
This sentence is really awkward and hard to read. Consider splitting into two.
Changed accordingly.

• Line 90-101: This entire paragraph is confusing. Are S and Cl mostly soluble like Na? How does increasing Na concentration lead to increased detection potential for S and Cl? It is not clear what you are trying to convey about the increase in Na concentration and ist relationship to other elements. Is Na increasing as a post-depositional process? Or is that a depositional thing for Na to be on the boundaries and that causes S and Cl to migrate as well?
This information needs to come earlier so that the reader is not reading for 4 pages wondering why we are looking at these specific elements.
The main point concerns the fact that localization at grain boundaries entails a local concentration increase with respect to what is measured in bulk liquid samples; several orders of magnitude are possible. We observed this already for Na (and other elements) (Bohleber et al., 2024) and this "high concentration at grain boundaries" effect may help us to detect Cl and S with LA-ICP-MS mapping, in spite of the analytical challenges connected to Cl and S.
We have clarified this further and moved it up in the introduction.

• Line 111-112: We thus focus on delivering information on the localization in the ice matrix over obtaining quantitative information on concentrations, which adds considerable complexity and uncertainty (Bohleber et al., 2024).

This concept comes out of nowhere. It would be helpful to add a sentence or two earlier in the introduction that presents this concept with more context. Why is it okay to just provide the spatial localization information? Do we need concentration data to successfully comment on the post-depositional diffusion hypotheses?

As observational microstructural impurity data, especially regarding S, remains scarce, we can already gain significant new insight on the diffusion mechanism from the localization of the impurities, hence concentrations are not needed (in this context, for now). We now mention this connection earlier in the introduction.

**Methods**

The methods section does not adequately explain how the study was conducted or why certain choices were made. This section needs to be revised to ensure that the methods are well explained.

First tell us about the samples you are going to work on. How you decided what to work on? Where are they from? What preparation steps did you have to go through to get the ice ready? Then, introduce the LAICPMS system(s) on which the samples were run. Tell us about how you ran them! Why did certain samples get run on one instrument vs. the other? Was there any post-processing of the data. If so, what steps did the data have to pass through? For the LA-ICP-MS section (currently 2.1), the authors need to clarify why they ran samples at two different labs. It is unclear to the reader if the authors were attempting to have an inter-lab comparison. If so, the results and discussions are confusing because they lack a section discussing the similarities or differences observed between the two runs for the one piece of ice – sample EDC 513 – run on both machines.

Section 2.2 "sample selection" should be greatly expanded to help the reader understand where the ice is from, why specific intervals were chosen, and what preparation had to happen before the samples were inserted into the cyrostage.

We agree with the referee that additional explanation will help the reader to understand the sample selection and combined use of two instruments at different laboratories. The data of the Graz laboratory came out of a previous investigation that was, initially, not primarily focused on sulfur. Yet this is where it first became clear that Cl and S can be mapped with LA-ICP-TOFMS. After the recent installation of the new LA-ICP-TOFMS at AWI, we have added further maps at higher resolution – and were again able to use a sample of EDC bag 513 (not the identical sample as in Graz, however). We have added text at the beginning of the method section and in "sample selection" to make this clearer.

We would further like to stress that this study was not designed as an inter-lab comparison, and have clarified on this. However, the fact that two labs generate qualitatively consistent data provides some form of validation. The combined use of data generated by different systems is not uncommon and we have done so previously for different purposes (Larkman et al., 2024), generally trying to exploit the individual strengths of each system.

The technical details have been described elsewhere already, namely concerning a) the LA-ICP-MS mapping method, map construction and decontamination protocol (Bohleber et al., 2020, 2021, 2023, 2024), b) the Graz LA-ICP-TOFMS system (Niehaus et al., 2024; Lockwood et al., 2024). Recently, we have added a respective publication on the new LA-ICP-TOFMS system at AWI (Bohleber et al., 2025). For the sake of brevity and readability of the manuscript, we are including a digest of the experimental steps involved. The interested reader can then consult the above referenced publications for more detail.

**Specific comments by line number:**

• Line 124: "Using a NIST612 glass standard ... distance between cell and sample surface."
Why not tune with a matrix matched ice standard?

The NIST glass has a homogenous distribution of elements, which facilitates tuning and methods development significantly. Using ice standards is here difficult as they will also form grain and boundaries with varying concentrations. Ablating these ice standards, we would not be able to benchmark instrument performance. We refer to Bohleber et al. (2024) for more details on the challenges of solid ice standards for LA-ICP-MS analyses and mention this now also in the text.

• Line 133: "The latter comprises an Analyte G2 laser ablation system (Teledyne Photon Machines, USA) equipped with a HelEx II ablation chamber compatible with the Venice cryostage."
You either need a citation for the Venice cryostage or a description here.

Added accordingly (Bohleber et al., 2020).

• Line 137-140: "Since using an identical ICP-TOFMS, the two systems are complementary through the differences in the laser ablation instrument."
Sentence is unclear – incorrect word choice with "through"? Why are you using two systems? An interlab comparison? Some other reason? How did you make the decision that one lab would run large spot sizes and large areas, and that the other lab would run small, high resolution maps? If I were to attempt to replicate your method, how would I know what spatial or resolution I should map at?

As explained above (and now in the text), the Graz analyses came first. Based on these results, we have been able to further advance the method with the system located at AWI providing higher resolution analysis with small spots. This new system enabled us to increase spatial resolution by conducting 10 and 1 micron spot analyses. The choice of resolution will always depend on the scientific question and the capabilities of the instrument, so no general answer is possible here. The chosen approach shows what you can expect from using two different systems - that the investigation of triple junctions greatly benefits from spot sizes as small as 1 micron which could be crucial for further analyses.

• Line 142: "Following our previously established approach for impurity mapping (Bohleber et al., 2020), the ice surface was decontaminated by scraping with a major-element free ceramic ZrO2 blade (American Cutting Edge, USA) immediately before inserting the samples into the ablation chamber."
This belongs in the sample selection and preparation sub-section.

Moved accordingly.

• Line 145: What were your pre-ablation methods? How many passes did you make? Did you record the optical mosaics before or after the pre-ablaiton?

As indicated by the reference in the text, this is already described in detail elsewhere (Bohleber et al., 2020) and has not changed. At least one preablation run is done with a larger spot size and low fluence. Optical mosaics are recorded after the measurement run.

• Line 146: "The maps were generated using HDIP (Teledyne Photon Machines)."
What is HDIP? Why do you need to use it to generate the maps? Is it something that is

done during or following the data collection?

Laser ablation set-ups don´t automatically produce chemical maps, but the laser position and chemical data have to be compiled into an "impurity map". For this crucial purpose, many commercial and open-source solutions exist. We are using the commercial HDIP software for final map generation after the data acquisition as explained in Bohleber et al. (2020). With the ICP-TOFMS you can also check the maps during data acquisition but post-measurement map processing is needed still.

• Line 151-156: "We chose a set of ice core samples both from the Holocene and the last glacial ... to provide a realistic subsample of typical polar ice conditions."
Can you give us more details about the cores when you introduce them here. Where exactly was it drilled. When was it drilled? Provide us with citations of other work that has been done on these cores. What time frame did you sample from? What parameters did you use to inform your sample selections? Was it purely limited by sample availability? Why did you choose these intervals of all the ice that exists? Explain not just that it is glacial, but why was *this* glacial sample selected? Are the samples from the inner part of the core? From the outer part of the core? What had to happen to the sample once you selected it? Did it have to be cut down to size? What was the cleaning procedure? Additionally, please explain why was only one sample run at both facilities? How was it determined which samples would be run where?

Referring to what is already explained above, we have added further detail to explain how the samples were chosen and prepared for analysis. The main motivation was in fact to look at samples with different chemical compositions, and therefore samples from the Holocene and the last glacial were selected. Numerous publications exist on the analysed ice cores and we thus refrain from adding basic information on them as it is not crucial to this manuscript dealing with specific analytes.

These samples were prepared similarly, with inner sections of the core cut specifically for LA-ICP-MS analysis, thus ensuring suitable dimensions for measurement (a cross section of 10 by 17 mm, with lengths between 5 cm and 9 cm, depending on core quality/breaks). These cuts were made in a cold room, with samples being placed immediately into clean vials after cutting, with pre-ablation in the LA-ICP-MS system providing suitable further decontamination.

• Line 153-154: "The glacial EGRIP sample contains a dust-rich cloudy band from the Younger Dryas and thus represents contrasting conditions to the low impurity samples from EDC."
You later refer to the samples with their sample number (i.e. EGRIP 2286). This system should be used throughout the manuscript for clarity.

Changed accordingly.

• Line 155: 250 000 particles/ml
Typo: 250,000 particles/mL

Changed accordingly.

**Results**
As currently written, the results and discussion sections are difficult to follow. Since The Cryosphere has no strict section requirements, consider if blending the results and discussion sections into a single section would help readability. Framing the section as:
To examine XXXX we made this map on this sample. We find XXXX. To better

understand XXXX, we analyzed the sample again higher resolution (?). To examine
this other hypothesis, we made a map of this sample XXXX, and found XXXX.
This would allow you to form a cohesive narrative that would benefit the reader's
comprehension of the science.

We understand why a single narrative might seem easier to follow first, but the manuscript
offers two different aspects of insight: a) the analytical challenge of mapping elements like Cl
and S in ice, revealing where they are located in the ice matrix, and b) the implication of
these findings for impurity diffusion studies. While there is certainly an element of personal
preference involved, we believe that it is clearer to treat Results and Discussion separately
under such circumstances. This allows us to present in detail the results acquired by the
different systems, and then to discuss them with respect to the two aspects a) and b). We
are optimistic that by addressing the comments from both referee reports, the revised
results and discussion sections will be clearer to the reader.

**Specific comments by line number:**
• Line 162-163: "Figures 1 and 2 show examples for the large size 32S intensity maps
alongside 37Cl and 23Na recorded with the Graz LA-ICP-TOFMS system."
This is the first time you have presented isotopes of elements. Since you will later go on to
show maps of multiple isotopes for each element, it would be good to add some text in the
methods explaining what isotopes you gathered and why it is helpful to look at more than
one.

Changed accordingly.

• Line 163: "Additional maps are included in the Supplementary Material"
Which maps are you referring to here? All of the supplementary material. There are no other
references to Figure S1 and Figure S3. Are they needed? If they are adding something
important, they should be discussed in further detail somewhere in the manuscript.

They are needed for the sake of showing all relevant data, but provide no added new insight
with respect to the main figures in the manuscript. Thus, we used the Supplementary
Material to display them.

• Line 163-165: "Na serves as a reference element showing the high degree of localization at
grain boundaries, analogous to previous observations in EDC and EGRIP (Bohleber et al.,
2020; Bohleber et al., 2023; Stoll et al., 2023).
This is a super important sentence and needs to be clearly stated in the introduction.

We agree on this as discussed above and have changed the text accordingly.

• Line 167-170: "The EGRIP cloudy band sample likely has significantly higher impurity
content compared to the EDC Holocene sample, reflected in the higher signal amplitude in
spite of the smaller spot size used (20 µm EGRIP vs 40 µm EDC)."
What is the sample number of the EDC sample? Why were the samples run at different spot
sizes? This needs to be explained with more clarity because it is not clear that comparing
different spot sizes is a good idea. Are you comparing apples and oranges, or is it okay to
compare like this?

It is the "EDC Holocene sample", hence EDC513 (added in the text). Different spot sizes were
used, especially in Graz, as they allow you to make a compromise between map area
covered and spatial resolution achieved in the same amount of time. The spot size also
affects the limits of detection (which is what is referred to here), and the impurity-rich EGRIP

sample showed clear signals of Cl and S even at 20 microns. For the comparatively lower impurity content in Antarctic Holocene ice, we chose a 40 micron spot. Arguably, an apples vs oranges situation would arise from a quantitative comparison, but here we are just noting that, in spite of using a smaller spot (less material ablated), the EGRIP sample still has higher intensities, consistent with the higher impurity concentration in Greenland glacial ice vs Antarctic interglacial ice. We have clarified this in the text.

• Line 183-187: "Using the AWI setup, maps were recorded on two subsamples ... the map on EDC513-1 (shown in the Supplementary Material)."
This should not be the first time that we are learning that EDC 1819 and EDC 513 were subsampled. This information should be provided in the "Sample Selection" portion of the methods. It should also be included in Table 2. It is unclear why it matters what order the samples were mapped in. If it is important, then the order of mapping should be included in the methods section. It would be helpful if you expanded the discussion of Figures 3-5. A brief description of each figure, including the sample, pixel size, and notable features, is warranted. What supplemental figure is EDC 513-1?
We agree that this detail was missing and have added information about the subsampling. Please note that this also has potential for further confusion (which we clear up in the revised text): 1819-1 and 1819-2 refer to two individual subsamples at adjacent depths, not an order of analysis (which does not matter).

• Line 189-190: "The map for 35Cl contains only noise in all maps recorded (hence not shown), but strong signal in 37Cl, which indicates the potential formation of Chlor- hydrogen adducts (see Sect. 4.1)."
It may be helpful to show both isotopes as a figure in the supplement. That way the reader doesn't have to trust you at your word. The discussion about Chlor-hydrogen adducts would be helpful to have right after you introduce this so the reader can follow easier.
We added one exemplary figure in the Supplement.

• Line 190: "As shown in Fig. 3 – 5 for S"
If you are not going to discuss each figure in detail, it is strange to have all three. Why show them if they are not noteworthy?
We are later referring to these Figures in the Discussion.

**Discussion**
**Specific comments by line number:**
• Line 223-225: "Although Na, S and Cl belong to the most abundant elements in polar ice (albeit generally at low ppb bulk concentrations)(Legrand & Mayewski, 1997), initial studies in LA-ICP-MS ice core analysis and in particular impurity mapping have focused on analytically more easily accessible elements such as Na, Mg, Al, Ca, Fe, Sr"
This idea is so crucial to the understanding of your research. This needs to be said *way* earlier, like in the introduction!! The text would also benefit from an additional prose about why Na, Mg, Al, Ca, Fe, and Sr are more accessible.
We have added text in the introduction in this regard. "More accessible" is meant here especially by comparison to S (interference) and Cl (low ionization).

• Line 226: "As demonstrated by the results shown in Fig. 1 – 7"
There are only six figures in the main text. Where is figure 7?

Well spotted. Should be Fig 1-6. Changed accordingly.

• Line 228-260: "The main hurdle for detecting S is the mass interference by ... collision-reaction cell gases, such as Xe or CH4 (Guillong et al., 2008; Singh et al., 2025)."
Both of these paragraphs need to be moved into the methods section when you talk about LA-ICP-TOFMS setup and analysis methodology.
Since this was not our method, or anything we have tried, we do not see this information in the method section but have included mentioning this in the introduction. Our rationale was to discuss the analytical challenge of measuring S and Cl in the discussion and to compare our technique to other approaches, like the ones mentioned here for S. Future studies may want to try these reaction gases for measuring S in ice with LA-ICP-MS if their ICP-MS allows it.

• Line 290: "the higher amount of resources consumed"
What kinds of resources? Ice? Or analytical resources (time, energy, gases)? You make a reference of 200k shots for the map in Figure 6, but the reader has no idea how many shots were needed for other maps. Therefore, it is hard to understand the increase in resources.
Primarily time, which also translates into consumption of resources like gases (Ar, He) and hence, not surprisingly, costs. Recent developments may help to find alternative strategies to the compromise between map size and spatial resolution (Larkman et al., 2025). We have reworded this sentence to make it clearer.

• Line 291: "Consequently, we only investigated this exemplarily in this work." There appears to be some missing text in this sentence.
Reworded.

• Line 294-295: "While at 10 µm spot size, grain boundaries are typically mapped with 2-3 high intensity pixels, for 5 µm this reduces to 1-2 pixels, with the same dosage 10 for all maps."
It is unclear how the number of high intensity pixels translates into data quality.
Not data quality per se but physical/chemical extent (width) of the grain boundaries is the point here. We have reworded the statement to point this out better.

• Line 298-299: "The 1 µm spot size may be more capable of resolving the fine-scale impurity variability around triple junctions, and shows that some enhancement at triple junctions relative to grain boundaries may exist at that spatial scale"
Enhancement of what? High intensity pixels?
Enhancement of Na at triple junctions relative to grain boundaries, shown as higher intensity pixels.

• Line 299-300: "In previous studies a comparable spatial resolution around 1x1 µm was used."
Citations needed.
Added accordingly.

• Line 324-327: "Accordingly, the physical reality likely is a mix between the two scenarios described in Ng (2021), with a highly localized nature of impurities in ice but variable

contributions from ice veins, grain boundaries and grain interiors depending on conditions, such as climate period and insoluble particle content, and uncharacterized microstructural processes."
This sentence is hard to understand. It would benefit from being split up into multiple sentences and more detail added.
Changed accordingly.

• Line 329-331: "Notably, the situation in deep ice sections, with larger grains and more time spent at higher temperatures, may deviate from the snapshots from relatively shallow depths presented here and remains to be explored, in particular regarding the influence of geochemical reactions leading to dissolution at grain boundaries (Baccolo et al., 2021)."
Why has this research been done at relatively shallow depth intervals, if developed methodology is supposed to be used on ultra-deep ice? Is it an ice availibility problem? How do you know that your method will still work on deeper ice core samples?
Two reasons: First, the diffusion studies also primarily focus so far on the last glacial cycle – as explained above. Second, the complex conditions in deep ice warrant a separate investigation – concerning large grains challenging the mapping from a practical point of view but more interesting, the suspected chemical reactions happening in deep ice. That said, there is no principal reason why this mapping should not work in deep ice, and deeper sections have been successfully mapped with LA-ICP-MS before (Bohleber et al., 2021; Stoll et al. 2023).

• Line 331-332: "Technically, the same is true for sections with high S-peaks from volcanic eruptions, which have not been measured here."
Please add citations to papers where the high S peaks have been measured, so the reader can learn more and see how your data compares.
This is likely a misunderstanding and needs to be clarified: To our knowledge, no studies on using LA-ICP-MS on ice containing volcanic S peaks have so far been published, but of course plenty of data from liquid analysis exists. We clarified this accordingly.

**References**
Many of the citations are incomplete, lacking dois and other information. In addition, your citations do not follow the format required by the journal. A complete list of reference requirements can be found at https://www.the-cryosphere.net/submission.html. Note that the year is at the end of the citation and that no part of the citation includes italicized text. Please work through the citations, fixing formatting and adding information where possible.
Changed accordingly.

• Here is the style for journal articles: Porter, J. G., De Bruyn, W., and Saltzman, E. S.: Eddy flux measurements of sulfur dioxide deposition to the sea surface, Atmos. Chem. Phys., 18, 15291–15305, https://doi.org/10.5194/acp-18-15291-2018, 2018.

• And the style for books: Singh, O. N. and Fabian, P. (Eds.): Atmospheric Ozone: a Millennium Issue, Copernicus Publications, Katlenburg-Lindau, Germany, 147 pp., ISBN 393658608X, 2003.

**Tables**
Table 1:

Consider reformatting this table so that it is easier to compare the laboratories with each other. This could be achieved in three columns: setting, value for AWI, and value for Graza. Why are the TOF settings not the same in the two labs? Is it because of the different lasers? What dictated changes to the rep rate?

The authors should have an expanded version of this table, either here or in the supplement, that lists all of the settings that were used. Note that these lists were generated for sample analysis on a Thermo Element 2, Thermo Element XR, and an Agilent 8900. Potentially, some settings are listed that are not appropriate for ICP-TOFMS or settings that need to be added to the list for ICP-TOFMS.

For the laser ablation system:
• Instrument Make/Model
• Software
• Ablation Cell
• Laser Wavelength
• Pulse Width
• RF Power
• RF Matching
• Sample Depth
• Gas Flow Setup (i.e. Direct connection)
• Carrier Gas Flow
• Energy Density
• Repetition Rate
• Spot Size
• Sampling Pattern (i.e. Single hole drilling or raster)
• Ablation Duration • Signal Smoother

For the ICP-TOFMS system:
• Instrument Make/Model
• Software
• Sample numbers measured
• RF Power
• Plasma Gas Flow
• Auxiliary Gas Flow
• Nebulizer Gas Flow
• Makeup Gas Flow
• Reaction Gas
• Reaction Gas Flow
• Analyzer Pressure
• Sample Inlet
• ARIS rapid aerosol introduction system settings
• Nebulizer make/model
• Sample uptake rate
• Sample Cone make/model
• Skimmer Cone make/model
• Torch Z position
• Sensitivity
• Scan Type

• Mass Window
• Integration Time or Window
• Detection Mode
• Isotopes Measured

We would rather present a concise overview of the two systems in Table 1 in the main text. The values are actually quite similar between the two systems and relevant differences are easily spottable and pointed out in the text (e.g. spot sizes). We also believe all relevant technical information – what is requested above- is already either in the table, the text or in the cited references. The scope of *The Cryosphere* is further not comparable to a methodological journal and these technical details would hamper readability without scientific value added.

Table 2:
This table should be expanded to include the laser spot size(s) used to help the reader understand what ice was measured with what settings. This is especially important because this information cannot easily be gleaned from Table 1. For audiences with broad backgrounds, consider adding a column noting if the sample is Holocene or Last Glacial. We have modified the table accordingly and point out the climatic period. We also decided to add further detail to the Figure caption directly (as also requested) and include detail on the mapping there, avoiding redundancies.

**Figures**
**General comments:**
Consider moving the element labels to the left y axis. This would allow you to stack the maps close together for ease of visual comparison. The scale bars would benefit from a white box behind it or to be moved off of the map. It is very difficult to read. Consider adding a scale bar to each image/map if you choose to leave them as is (not closely stacked). Does cts = counts per second (cps)? Cts should be explained. Consider adding more details to the figure caption. Talk about the sample time frame that we are looking at and any notable features that the reader should focus on. Please reference the specific figures in the supplement instead of just saying "Supplementary Material".
We expanded the figure caption as requested. Explained now that "cts" is counts, not counts per second – albeit this differs only by a constant factor.

**Specific Comments:**
Figure 2: Make sure that your optical mosaic is cropped to the same extent as the LA-ICP-TOFMS elemental maps.
This is a marginal cut caused by a different spatial resolution of the camera compared to the laser spot size, but done anyway.

Figure 6: It seems that this figure would work better as a 3 panel strip; in each row – optical, 10um, 5um, 1um for a different element/isotope. If you keep it as is, please add an element/isotope label to the 5 um and 10 um spot size panels.
Changed accordingly

Figure S2: Is it possible to adjust the contrast on the mosaic image to improve the visibility of the grain boundaries?

It remains challenging, but we have tried to further improve the contrast as can be seen in the modified manuscript.

Figure S3: The intensity is now in au. Why the shift from cts? What is au?
We changed this to cts to be consistent.

References

- Barnes, P. R. F., Wolff, E. W., Mader, H. M., Udisti, R., Castellano, E., & Röthlisberger, R. (2003). Evolution of chemical peak shapes in the Dome C, Antarctica, ice core. Journal of Geophysical Research: Atmospheres, 108(D3).
- Bohleber, P., Mervič, K., Dallmayr, R., Stremtan, C., & Šala, M. (2025). Argon versus helium as carrier gas for LA-ICP-MS impurity mapping on ice cores. Talanta Open, 11, 100437.
- Bohleber, P., Larkman, P., Stoll, N., Clases, D., Gonzalez de Vega, R., Šala, M., ... & Barbante, C. (2024). Quantitative insights on impurities in ice cores at the micro-scale from calibrated LA-ICP-MS imaging. Geochemistry, Geophysics, Geosystems, 25(4), e2023GC011425.
- Bohleber, P., Stoll, N., Rittner, M., Roman, M., Weikusat, I., & Barbante, C. (2023). Geochemical Characterization of Insoluble Particle Clusters in Ice Cores Using Two-dimensional Impurity Imaging. Geochemistry, Geophysics, Geosystems, 24(2), e2022GC010595.
- Bohleber, P., Roman, M., Stoll, N., Bussweiler, Y., & Rittner, M. (2021). Imaging the distribution of elements in antarctic ice cores with LA-ICP-TOFMS. TOFWERK Application Note.
- Bohleber, P., Roman, M., Šala, M., & Barbante, C. (2020). Imaging the impurity distribution in glacier ice cores with LA-ICP-MS. Journal of Analytical Atomic Spectrometry, 35(10), 2204–2212.
- Fischer, H., Severinghaus, J., Brook, E., Wolff, E., Albert, M., Alemany, O., ... & Wilhelms, F. (2013). Where to find 1.5 million yr old ice for the IPICS" Oldest-Ice" ice core. Climate of the Past, 9(6), 2489-2505.
- Larkman, P., Vascon, S., Šala, M., Stoll, N., Barbante, C., & Bohleber, P. (2025). Faster chemical mapping assisted by computer vision: insights from glass and ice core samples. Analyst.
- Larkman, P., Rhodes, R. H., Stoll, N., Barbante, C., & Bohleber, P. (2024). What does the impurity variability at the microscale represent in ice cores? Insights from a conceptual approach. EGUsphere, 2024, 1-25.
- Lockwood, T. E., De Vega, R. G., Du, Z., Schlatt, L., Xu, X., & Clases, D. (2024). Strategies to enhance figures of merit in ICP-ToF-MS. *Journal of Analytical Atomic Spectrometry*, *39*(1), 227-234.
- Niehaus, P., de Vega, R. G., Haindl, M. T., Birkl, C., Leoni, M., Birkl-Toeglhofer, A. M., ... & Clases, D. (2024). Multimodal analytical tools for the molecular and elemental characterisation of lesions in brain tissue of multiple sclerosis patients. *Talanta*, *270*, 125518.
- Stoll, Nicolas, Eichler, J., Hörhold, M., Shigeyama, W., & Weikusat, I. (2021). A review of the microstructural location of impurities in polar ice and their impacts on deformation. *Frontiers in Earth Science*, *8*, 658
- Stoll, N., Westhoff, J., Bohleber, P., Svensson, A., Dahl-Jensen, D., Barbante, C., and Weikusat, I. (2023). Chemical and visual characterisation of EGRIP glacial ice and cloudy bands within, The Cryosphere, 17, 2021–2043, https://doi.org/10.5194/tc-17-2021-2023.
- Wolff, E. W., Fischer, H., van Ommen, T., & Hodell, D. A. (2022). Stratigraphic templates for ice core records of the past 1.5 Myr. Climate of the Past, 18(7), 1563-1577.

---

## Author Response (AR2)

**Revision of EGUSPHERE2025-355**

Revision and response to comments by editor Kaitlin Keegan – in blue

Dear Dr. Bohleber and co-authors,

Thank you for your revised manuscript based on the referees' comments. I agree that the readability of both the figures and the text of the manuscript has been improved. Below, I've included a list of specific comments that would further enhance the clarity of the writing. Regarding the reviewer's suggestion to combine the results and discussion sections, I agree with you and your preference to keep the sections separate. To make the structure of the results section clearer, consider either indicating the importance of looking at the samples in those three ways (sample sizes and measurement resolutions) in a previous section or renaming the results subsections based on the key finding or focus instead of the sample size or measurement resolution.

Thank you very much for these additional comments that helped us to further improve the clarity of the manuscript. We have changed the manuscript accordingly in all instances. Further details are provided below.

We have added an introductory paragraph to the results sections to better explain to the reader the rationale behind the subsections. We agree that this way, the structure of the result section should become clearer to the reader.

The following line comments are based on the line numbers in the tracked-changes version of the manuscript:

L18: consider rephrasing to something like "In ice without evidence of volcanic activity, and unenhanced impurity concentrations, we obtain..."

**Changed accordingly.**

L19-20: 'exemplarily even higher resolution maps..." is awkward. Consider rephrasing to something like: "... and also include some exemplarily high resolution maps with a spot size down to 1 micrometer."

**Changed accordingly.**

L20: It's awkward to state "We use Na as a previously investigated reference element..." and not describe your Na findings here, because you describe your findings of S and Cl in the second half of the sentence.

**Changed accordingly.**

L24: should be "These results..."

**Changed accordingly.**

L56: should be "In the case of sodium,..."

We mean here that the case of sodium, chloride and sulfate makes up an important reference ensemble...we have removed "the case of" altogether.

L59: add 'of' after 'diffusion'

**Changed accordingly.**

L143: a word is missing in the phrase "... datasets that allow to draw generalized conclusions." Change to something like "...datasets that allow us to draw generalized conclusions." or "...datasets that allow for drawing generalized conclusions."

**Changed accordingly.**

L182: stay consistent with how you refer to the impurities you focus on in this study. It

seems appropriate to use their common names in the Abstract and the first time they're mentioned within the main text of the manuscript, and then use their symbols for subsequent references to them.

We understand this comment. Our reasoning was that in the few sentences that refer to "sodium, chloride and sulfate" we would spell out the names of the ionic impurities (instead of using  $Na^+$ ,  $Cl^-$ , and  $SO_4^{2-}$ ) – and strictly use the symbols for the elemental impurities that are measured with ICP-MS. We have now added "ionic" to these sentences where needed in order to make this more evident.

L186: it's not clear what you're referring to in the phrase "within the range of this technique" Clarified accordingly.

L189: Something like the following would be a clearer sentence to add here: "Na serves as a reference element to compare our maps to previous LA-ICP-MS maps."

Changed accordingly.

L195-196: by "to decide on", I think you mean something like: "Note that localization is key to determining the plausability of all these scenarios, and a fully quantitative analysis is not required."

Changed accordingly.

L197: "rather than" instead of "over" here would make the sentence clearer.

Changed accordingly.

Figure 1: Is the bottom left panel missing the scale bar?

Changed accordingly.

L524: this phrase is awkward and missing some words to make it clear: "... the maps visible if changing the applied resolution."

Changed accordingly.

L525: this phrase is awkward and missing some words to make it clear: "...the same dosage 10 for all maps."

Changed accordingly.

L561: "showed" should be "shown"

Changed accordingly.

L562: "bands" should be singular

Changed accordingly.

L563: it's not clear what the phrase "less abundant cases" means here; "present also" should be "also present"

Clarified accordingly.

L564: I think that you mean "On these grounds" here

Changed accordingly.

L609: "due to" instead of "by" would make this sentence clearer

Changed accordingly.

L611: the correct phrasing is either "resulting from" or "due to"

Changed accordingly.

L623: consider adding a phrase like "potential explanations" or "potential mechanisms" before "i) and ii)" to make this statement clearer

Changed accordingly.

L626: "large" instead of "high"

Changed accordingly.

L627: "practicality" instead of "practicability"

**Changed accordingly.**

L627-628: the phrase "... but similar to the 1 micron mapping exemplary investigations could be in reach soon." is confusing. Consider rewriting this for clarity.

**Changed accordingly.**

L631: "found" instead of "find"

**Changed accordingly.**

L632: "constraining better" is an awkward phrase, consider rewording to something like "constraining the mobility..." or "more precisely constraining the mobility..."

**Changed accordingly.**

L642: the colon should be a period; add a comma after "maps"

**Changed accordingly.**

L643: add "and" before "hence"

Changed accordingly.

L663: add "at" before "1"

Changed accordingly.

All the best, Kaitlin